# A dynamic rhizosphere interplay between tree roots and soil bacteria under drought stress

Yaara Oppenheimer-Shaanan, Gilad Jakoby, Maya L Starr, Romiel Karliner, Gal Eilon, Maxim Itkin, Sergey Malitsky, Tamir Klein*

Department of Plant and Environmental Sciences, Weizmann Institute of Science, Rehovot, Israel

**Abstract** Root exudates are thought to play an important role in plant-microbial interactions. In return for nutrition, soil bacteria can increase the bioavailability of soil nutrients. However, root exudates typically decrease in situations such as drought, calling into question the efficacy of solvation and bacteria-dependent mineral uptake in such stress. Here, we tested the hypothesis of exudate-driven microbial priming on *Cupressus* saplings grown in forest soil in custom-made rhizotron boxes. A 1-month imposed drought and concomitant inoculations with a mix of *Bacillus subtilis* and *Pseudomonas stutzeri*, bacteria species isolated from the forest soil, were applied using factorial design. Direct bacteria counts and visualization by confocal microscopy showed that both bacteria associated with *Cupressus* roots. Interestingly, root exudation rates increased 2.3-fold with bacteria under drought, as well as irrigation. Forty-four metabolites in exudates were significantly different in concentration between irrigated and drought trees, including phenolic acid compounds and quinate. When adding these metabolites as carbon and nitrogen sources to bacterial cultures of both bacterial species, eight of nine metabolites stimulated bacterial growth. Importantly, soil phosphorous bioavailability was maintained only in inoculated trees, mitigating drought-induced decrease in leaf phosphorus and iron. Our observations of increased root exudation rate when drought and inoculation regimes were combined support the idea of root recruitment of beneficial bacteria, especially under water stress.

## Editor's evaluation

This paper will be of interest to those interested in plant-microbe interactions under drought. Trees can exchange root exudates for minerals with soil bacteria. In a pot experiment under realistic conditions, the authors indicate that this exchange persists, and may be protective, when plants experience drought stress. The combination of methods used is an important strength: visualizing bacterial colonization of roots, measuring root exudation and mineral uptake, in conjunction with separate assays of bacterial growth responses to chemicals released by trees.

## Introduction

Climate change is characterized by increased temperatures and altered precipitation patterns (*Cavin et al., 2013*; *IPCC, 2014*). These environmental changes affect terrestrial ecosystems worldwide, with negative impacts for forest health associated with water limitation (*Zhao and Running, 2010*; *Allen et al., 2010*; *Choat et al., 2012*). Drought adversely affects forest health in many aspects, including seedling recruitment (*Pozner et al., 2022*), tree productivity (*Klein et al., 2014*), and mortality of trees (*Klein et al., 2019*), with increased susceptibility to pathogen or insect attack (*Reichstein et al.,*

**\*For correspondence:**
tamir.klein@weizmann.ac.il

**Competing interest:** The authors declare that no competing interests exist.

**eLife digest** The soil surrounding the roots of trees, termed the rhizosphere, is full of bacteria and other communities of microorganisms. Trees secrete organic compounds in to the soil which are thought to influence the behavior of bacteria in the rhizosphere. Specifically, these root secretions, or 'exudates', attract and feed soil bacteria, which, in return, release nutrients that benefit the tree.

In 2020, a group of researchers found that some trees in the Mediterranean forest produce more exudates during the long dry season. This suggests that the compounds secreted by roots may help trees to tolerate stress conditions, such as drought. To test this hypothesis, Oppenheimer-Shaanan et al. – including some of the researchers involved in the 2020 study – grew young *Cupressus sempervirens* conifer trees in drought conditions that starved them of the nutrients phosphorous and iron.

Each tree was planted in a custom-built box which allowed easy access to roots growing in the soil. Two species of bacteria from the forest soil C. *sempervirens* trees naturally live in were then added to the soil in each box. Microscopy revealed that both species of bacteria, which had been tagged with fluorescent markers, were attracted to the roots of the trees, boosting the bacterial community in the rhizosphere.

Oppenheimer-Shaanan et al. found that the recruitment of the two bacterial species caused the rate at which exudates were secreted from the roots to increase. Compounds in the exudate stimulated the bacteria to grow. Ultimately, levels of phosphorous and iron in the leaves of the starved trees increased when in the presence of these soil bacteria. This suggests that bacteria in the rhizosphere helps trees to survive when they are under stress and have low levels of water.

These findings provide further evidence that plants and bacteria can live together in symbiosis and benefit one another. This could have important implications for forest ecology and potentially how trees are grown in orchards and gardens. For example, specific bacteria and organic compounds in the rhizosphere may be able to improve tree health. However, further work is needed to investigate whether the exudate compounds identified in this study are found more widely in nature.

---

*2013*; *McDowell et al., 2022*). Moreover, both drought frequency and intensity are projected to increase; particularly in the Mediterranean region and southern Africa (*IPCC, 2019*). The capacity of trees to survive future droughts is virtually unknown (*McDowell et al., 2022*).

Trees have evolved mechanisms to cope with drought, including escape, avoidance, and tolerance strategies. Drought has been shown to induce an alteration of carbon allocation from aboveground to belowground organs and increase in the amounts of soluble sugars in the roots (*Huang and Fu, 2000*; *Hasibeder et al., 2015*). Root systems mediate water and nutrient uptake, provide physical stabilization, store nutrients and carbohydrates, and provide carbon and nutrients to the soil through the process of fine-root turnover (*Brunner and Godbold, 2007*; *Haichar et al., 2008*; *Ryan, 2011*; *Harfouche et al., 2014*; *Jarzyniak and Jasiński, 2014*; *Klein et al., 2016*). In addition to these roles, trees invest a substantial part of their photosynthesized carbon into root exudates that entice and presumably feed plant-beneficial and root-associated microbiota (*Bais et al., 2006*; *Badri and Vivanco, 2009*; *Karst et al., 2017*; *Jakoby et al., 2020*). In parallel, the rhizosphere microbes can promote plant growth through various mechanisms such as increasing the availability of nutrients, secreting phytohormones, suppressing pathogens, or having positive effects on the plant metabolism (*Pérez-Montaño et al., 2014*; *Zhou et al., 2016*). Microbial utilization and metabolism play a central role in modulating concentration gradients of a variety of compounds right outside root tips, thereby constituting a soil sink (*Dakora and Phillips, 2002*; *Mommer et al., 2016*; *Martin et al., 2017*; *Tsunoda and van Dam, 2017*). The majority of root exudates typically consists of primary metabolites (20%; sugars, amino acids, and organic acids) and 15% of nitrogen as well as secondary metabolites, complex polymers, such as flavonoids, glucosinolates, auxins, etc. (*Vives-Peris et al., 2020*). Those plant-derived metabolites were shown to shape microbial communities by allowing bacteria to metabolize them and then establish themselves in the rhizosphere (*Venturi and Keel, 2016*; *Sasse et al., 2017*). Although root exudation is ubiquitous among tree species, the amount and composition of root exudates vary. So far, little information is available on how drought influences tree root exudates, their chemical composition, and how root metabolism is connected with shifts in root-associated microbiome

composition (*Tückmantel et al., 2017*; *Xu et al., 2018*). Recently, it was shown that oak trees (*Quercus ilex*) shift their exudates from primary to secondary metabolites under drought (*Gargallo-Garriga et al., 2018*).

The chemical composition of root exudates have a direct effect on the rhizosphere communities. These include plant growth-promoting rhizobacteria (PGPR) genera such as *Bacillus, Pseudomonas, Enterobacter, Acinetobacter, Burkholderia, Arthrobacter,* and *Paenibacillus* (*Sasse et al., 2017*; *Zhang et al., 2017*). For example, the banana root exudate fumaric acid attracts the Gram-positive *Bacillus subtilis* N11 and stimulates biofilm formation (*Zhang et al., 2014*); malic acid exuded by *Arabidopsis* stimulates binding to roots and biofilm formation on roots by the *B. subtilis* strain FB17 (*Rudrappa et al., 2008*). Bacterial growth and antifungal activity of certain species of the Gram-negative *Pseudomonas* spp. is dependent on organic acids and sugars isolated from tomato root exudates (*Kravchenko et al., 2003*; *Mhlongo et al., 2018*). Among PGPR, species of *Pseudomonas* and *Bacillus* are the best studied as models for beneficial plant-microbe interaction (*Weller et al., 2002*; *Raaijmakers et al., 2010*). Interestingly, the importance of both genera to plants has been verified in multiple metagenomic studies (*Loper et al., 2012*; *Mendes et al., 2013*), however, the interplay between these important root bacteria and forest trees remains a mystery. In the forest soil, root exudation is suspected to enhance rhizobacteria, in turn leading to increased decomposition of soil organic matter, that is, increased C mineralization, a process termed microbial priming (*Schleppi et al., 2019*). Here, we use this term on a wide perspective, even that P release from soil phosphates, for example, is not strictly considered as priming (*Dijkstra et al., 2013*).

Recently, we have shown that roots of both conifer and broadleaf tree species in an evergreen Mediterranean forest increase their exudation flux during the long dry season (*Jakoby et al., 2020*). This increase in exudation occurred in spite of the sharp decrease in photosynthesis throughout the dry season, and more so in the coniferous *Cupressus sempervirens*. Hence, we speculated a specific role for root exudation in tree drought tolerance. Although the interactions between the rhizosphere microbiome and plants play a crucial role in plant growth, it is still unclear how these interactions affect trees under abiotic stresses. Our overall objective was to test whether microbial recruitment by tree root exudates (a form of microbial priming) is beneficial to trees under drought, an abiotic stress that alters tree carbon and nutrient allocation. Here, we link together rhizosphere processes and drought by studying the effect of soil drought on the physical and chemical interaction between the rhizosphere bacteria *B. subtilis* and *P. stutzeri* and roots of the conifer species *C. sempervirens*. We designed a custom-made tree growth facility permitting access to intact roots growing in forest soil, where we measured root exudation rate and composition in response to changes in irrigation and inoculations of these soil bacteria (*Figure 1—figure supplement 1*). Our major hypothesis regarded the existence of an exudate-induced microbial-tree interaction cycle, starting with tree stress and subsequent exudation, on to enhancement of soil bacteria and their activity, and back to improved tree nutrition.

## Results

### Soil drought limits tree gas exchange and growth

Soil moisture declined gradually in drought trees down to <10% (V/V) 3 weeks following irrigation cessation, and increased back after re-irrigation (*Figure 1—figure supplement 2*). In the beginning of the manipulation, which followed the end of the wet season (May 2019), we needed to adjust the irrigation amounts for the irrigated trees, which stabilized within 11 days. Assimilation and stomatal conductance of watered trees fluctuated around 8–12 $\mu$mol $CO_2$ $m^{-2}$ $s^{-1}$ and 150–300 mol $H_2O$ $m^{-2}$ $s^{-1}$ throughout the experiment, respectively (*Figure 2—figure supplement 1*; *Figure 2—source data 1*). In drought trees, these high rates decreased to zero within 4 weeks, and, upon re-irrigation, increased back to baseline values within 4–5 weeks. Bacterial inoculations had no effect on leaf gas exchange, and hence the interaction irrigation:bacteria was not significant either. During the experiment, sapling biomass increment was 188.6±15.6 and 149.1±14.4 g for irrigated saplings with and without bacterial inoculations, respectively (difference not significant; p=0.753; *Figure 2—source data 1*). The effect of drought was highly significant (p<0.001), and biomass increment in drought-exposed saplings with and without bacterial inoculations was 22.4±5.7 and 4.5±4.6 g, respectively (p=0.09).

## Trees recruit root-associated bacteria during drought

Fluorescently tagged *B. subtilis* and *P. stutzeri*, that were modified from native strains isolated from the forest soil, showed attachment and dispersed colonization along *Cupressus* fine roots, regardless of irrigation, on days 1 and 3 following inoculation (*Figure 1—source data 1*). Bacteria seemed to localize along crevices in the root epidermis, forming lines along roots, and sometimes co-localized (*Figure 1A*). The inoculations can be regarded as pulses, allowing us to test the rhizosphere interactions, while declining with time. Upon inoculation, *P. stutzeri* had higher abundance (~90%) than *B. subtilis*, but this ratio reversed within a few days or weeks (*Figure 1B*), with *B. subtilis* having an advantage over *P. stutzeri* in drought trees. Interestingly, when irrigation was restored, the drought trees recruited the bacteria in a similar fashion to irrigated trees, with *P. stutzeri* exhibiting 80% abundance in the rhizosphere at day 3, and then substituted by *B. subtilis* at days 7 and 15 (*Figure 1C*). Differences between days in relative abundance were significant (p<0.001; *Figure 1—source data 1*), except for the difference between days 3 and 15 in soil under drought. Bacterial abundance decreased over time, but, importantly, was always ~10-fold higher in the rhizosphere than in the bulk soil (*Figure 1B*), further evidencing recruitment by roots. Bacterial abundance was also lower in drought trees than around irrigated (*Figure 1B*) or re-irrigated trees (*Figure 1C*) (p<0.001; *Figure 1—source data 1*). Overall, only in trees under constant irrigation and after two inoculations, bacterial communities of *B. subtilis* and *P. stutzeri* seemed to stabilize in the rhizosphere (*Figure 1C*).

## Tree root exudates respond to drought and bacterial inoculation

Under constant irrigation and under drought, bacterial inoculation significantly increased root exudation, by two- to threefold (*Figure 2*; p<0.001). However, in trees that were exposed to drought and then re-irrigated, this pattern reversed, and inoculated trees had slightly lower exudates than without bacteria. Thus, drought-exposed trees, that were responsive to inoculation, showing an ~50% increase in root exudation, lost this response when re-irrigated (p=0.039 for the interaction irrigation:bacteria). The amount of organic carbon exuded from roots of drought-exposed trees was significantly, ~50% lower than from roots of irrigated trees, and moreover, ~70% lower, in the re-irrigation period (across inoculation treatments; p<0.01; *Figure 2—source data 2*). Importantly, the same trees that were exposed to drought reduced their exudation when re-irrigated (*Figure 2*). This observation is further supported by examining correlation coefficients between root exudation rates and the abundance of *B. subtilis* and *P. stutzeri* in the rhizosphere. They yielded values above 0.5 for drought trees on day 3 following inoculation (*Figure 2C*; p=0.046 for *P. stutzeri*). For *B. subtilis*, there was a high correlation also with irrigated trees' exudation on day 7 following inoculation (p=0.093). Correlation coefficients decreased in the re-irrigation period, and were negative in trees that were constantly irrigated. An additional measure of tree carbon allocation into root exudation is the ratio between exudation rate (μg C mg root$^{-1}$ day$^{-1}$) and net assimilation rate (μmol C m$^{-2}$ leaf s$^{-1}$). This ratio was 0.40–0.45 under drought, decreasing to 0.27–0.35 under re-irrigation (across inoculation treatments). In irrigated saplings, ratios increased from 0.12 to 0.17 without bacteria, to 0.34 and 0.58 in inoculated saplings (in later and earlier phases of the experiment, respectively).

## Rhizosphere bacteria induce systemic changes in root exudate composition

Roots of drought-exposed trees displayed substantial changes in exudate composition compared to irrigated trees, and more so when inoculated with rhizosphere bacteria (*Figure 3—source data 1*). Principal coordinate analysis (PCoA) showed that root exudates blends (samples) from drought trees that were inoculated and those from drought trees that were not inoculated partitioned into distinct clusters, with partial overlap (*Figure 3A*). When samples from irrigated trees were also considered in the PCoA, they behaved similarly, however there was no single sample common to all three groups (*Figure 3A*). Metabolomics data identified 149 masses (52.6% of the total mass entities) that were significantly different from either *Cupressus* root tissues or a root-free control solution incubated in the soil, and were hence regarded as *Cupressus* exudates (*Figure 3—source data 1*). About 30 of these metabolites were 2- to 10-fold more abundant in the exudates blends when compared to the controls, and the rest were 15- to 25-fold more abundant (*Figure 3B*, *Figure 3—source data 1*). Of the latter number, 44 metabolites were significantly different between irrigated and drought trees (p<0.05; *Figure 3—source data 1*), and 17 metabolites were significantly different between irrigated and

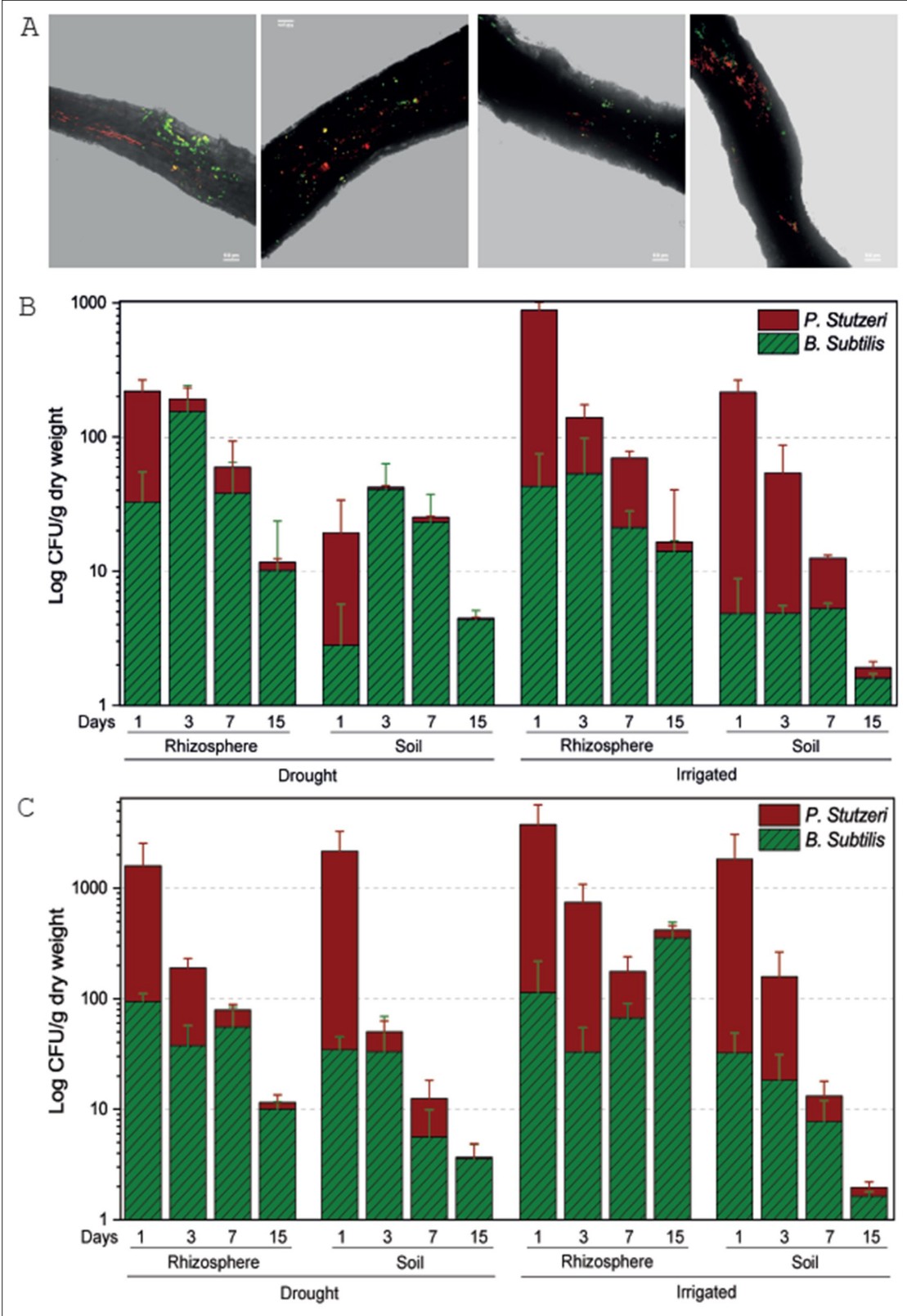

**Figure 1.** Tree root recruitment of beneficial bacteria during drought and re-irrigation. Bacterial dynamics in soil and rhizosphere around roots of irrigated and drought-exposed *Cupressus sempervirens* saplings. Root colonization (**A**): epifluorescence and bright-field images of drought tree roots densely colonized by *Bacillus subtilis*-gfp (green) and *Pseudomonas stutzeri*-mCherry (red). Orthogonal views of a three-dimensional confocal image were created from a z-stack of x/y-scans on drought trees, 1 and 3 days following inoculation. Dynamics of relative abundance of *B. subtilis* (green)

*Figure 1 continued on next page*

*Figure 1 continued*

and *P. stutzeri* (red) in rhizosphere and soil during drought (**B**) and during re-irrigation (**C**). Presented are colony forming units of both species (CFU; expressed as log 10 per g root dry weight) (n=6).

The online version of this article includes the following source data and figure supplement(s) for figure 1:

**Source data 1.** Statistical analysis of bacterial growth.

**Figure supplement 1.** Experimental system and climate.

**Figure supplement 2.** Changes in soil water content (% v/v) in irrigated (blue) and drought-exposed (orange) saplings along the course of the experiment.

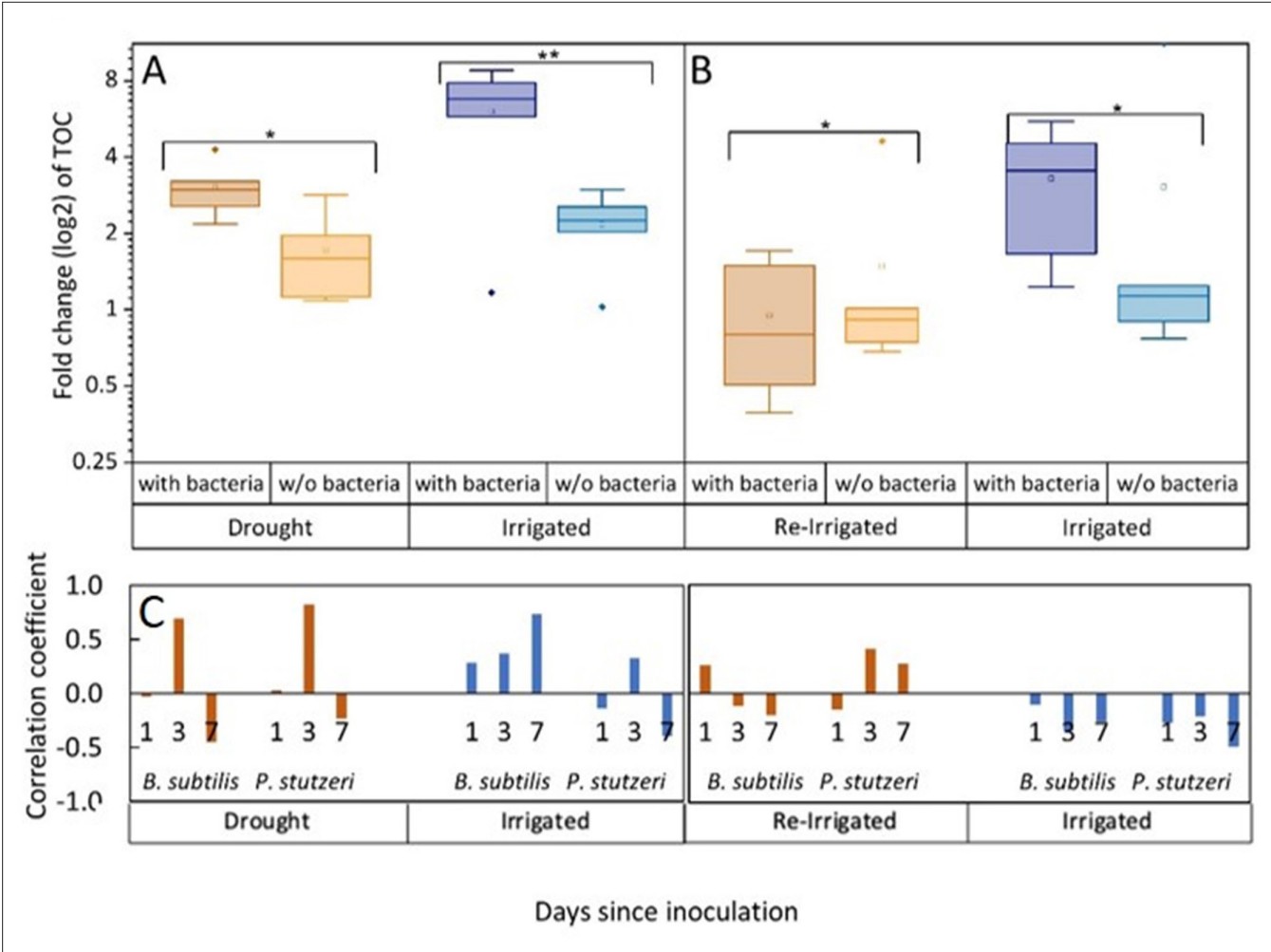

**Figure 2.** Tree root exudates increase with bacterial inoculation for both the drought and irrigation treatments (**A**) and decrease with bacterial inoculation after rewetting the droughted trees (**B**). Total organic carbon (TOC) in exudate solutions from roots of irrigated and drought-exposed *Cupressus sempervirens* saplings, with and without bacterial inoculations. Intact roots were incubated for 48 hr to collect exudates during periods of drought (**A**) and re-irrigation (**B**). Boxplots show the log 2 of fold change from baseline exudation rate (at the beginning of the experiment) in µg C mg root$^{-1}$ day$^{-1}$. Asterisks indicate significant differences based on two-way ANOVA performed with Tukey's HSD test (n=6, p<0.05) (see *Figure 2—source data 2*). Coefficients for the correlations between exudate rate (TOC) and rhizosphere abundance of *Bacillus subtilis* or *Pseudomonas stutzeri* (as in *Figure 1*) for the specific tree groups at 1, 3, and 7 days following inoculation (**C**).

The online version of this article includes the following source data and figure supplement(s) for figure 2:

**Source data 1.** Statistical analysis of gas exchange parameters and sapling biomass.

**Source data 2.** Statistical analysis of root exudates total organic carbon.

**Figure supplement 1.** Drought stress effect on tree physiological parameters.

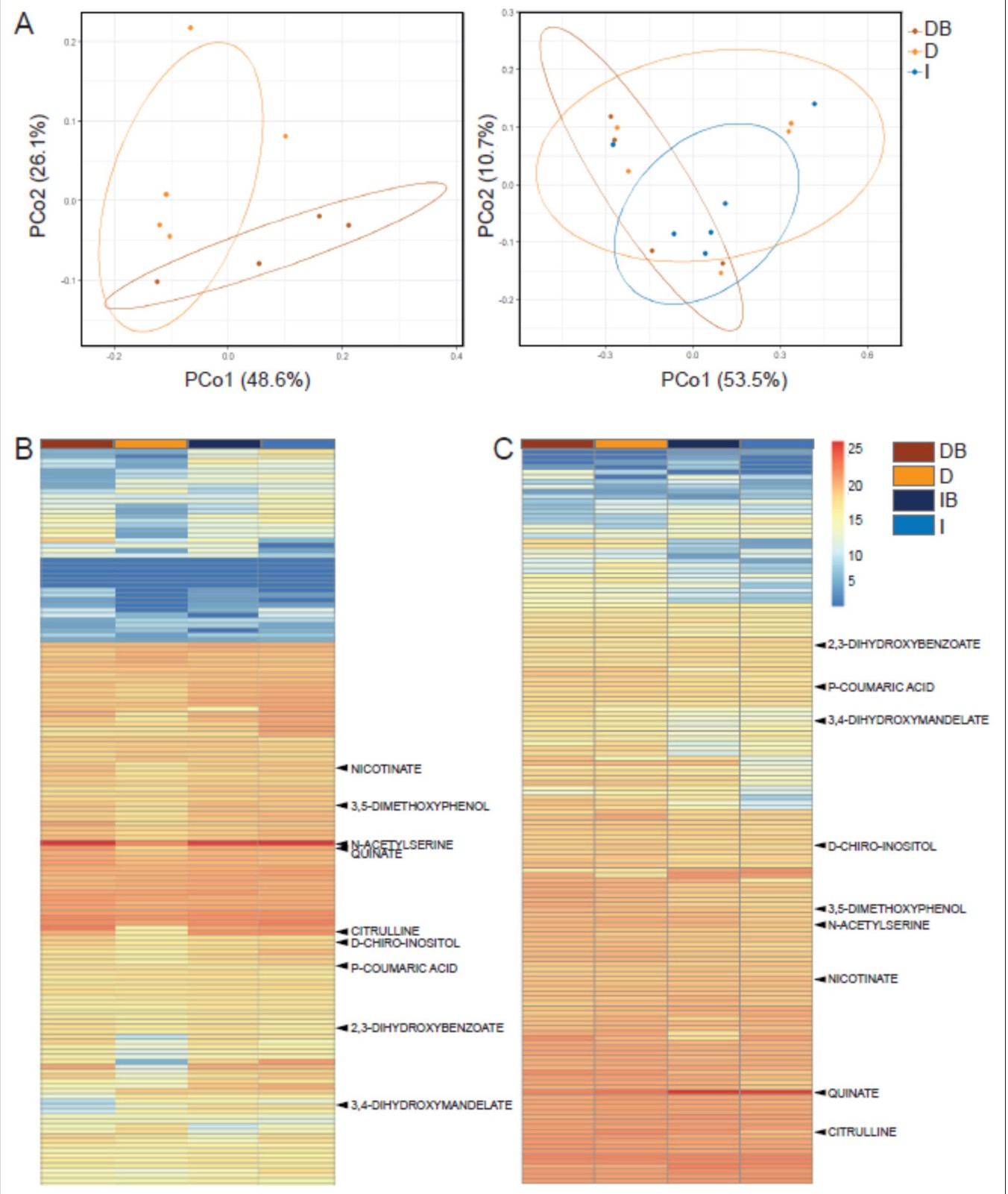

**Figure 3.** Rhizosphere bacteria and drought induce systemic metabolic changes in root exudates. Metabolic profiles of root exudate solutions from roots of irrigated and drought-exposed *Cupressus sempervirens* saplings, with and without bacterial inoculations. Intact roots were incubated for 48 hr to collect exudates, which were analyzed by mass spectrometry. (DB) drought with bacteria, (D) drought w/o bacteria, (IB) irrigated with bacteria, (I) irrigated w/o bacteria. (A) Principal coordinate analysis of the polar metabolite profiles of exudates from drought-exposed saplings with and without

*Figure 3 continued on next page*

*Figure 3 continued*

bacterial inoculations (DB and D, respectively; left), and in combination with irrigated saplings (I; right; excluding the group of irrigated trees with bacteria for clarity). (**B**) Heat map analysis of the fold change of each of 149 identified metabolites (rows) relative to control level (at the beginning of the experiment). Columns represent means for each of the four saplings groups (n=4–6). (**C**) Same as in (**B**) for the re-irrigation phase. The nine metabolite names indicate metabolites which were tested in vitro for bacterial growth. For the list of metabolites, see *Figure 3—source data 1*.

The online version of this article includes the following source data and figure supplement(s) for figure 3:

**Source data 1.** Data and statistical analysis of root exudates polar metabolites.

**Source data 2.** Statistical analysis of root exudates semi-polar metabolites.

**Figure supplement 1.** Semi-polar metabolic profiles of root exudate solutions from roots of irrigated and drought-exposed *Cupressus sempervirens* saplings, with and without bacterial inoculations.

re-irrigated trees (p<0.05) (*Figure 3—source data 1*). In addition, 13 metabolites were significantly enriched or depleted following bacterial inoculation under drought and one metabolite changed following bacterial inoculation after re-irrigation (*Figure 3—source data 1*). In two metabolites, the interaction irrigation:bacteria was significant under drought, but not under re-irrigation (*Figure 3— source data 1*). The metabolites that differed between treatments represented compounds of diverse chemistries, including hydroxycinnamic acid conjugates, hydroxybenzoic acid derivatives, organic acid derivatives, purine derivatives and pyrimidine derivatives, amino acids, and amino acid metabolism derivatives, sugars, and sugar derivatives (*Figure 3—source data 1*). During drought, many metabolites were depleted compared to their abundance under irrigation, however, upon bacterial inoculation, their exudation was enriched again (*Figure 3B*). During re-irrigation, many metabolites were enriched compared to their abundance under constant irrigation, and bacterial inoculation did not change the metabolite profile (*Figure 3C*). Importantly, the exudates in drought conditions consisted mainly of secondary metabolites (70% of total metabolites) associated with tree responses to drought stress (*Figure 3—source data 1E*). In contrast, the metabolite composition under re-irrigation shifted toward the dominance of primary metabolites (85% of total metabolites), and differences compared to irrigated trees were not significant.

In addition to the analysis of polar compounds, an untargeted semi-polar metabolite profiling approach yielded 2285 clustered mass signals which were annotated as semi-polar putative compounds during drought. PCA (principal component analysis) of exuded semi-polar metabolites identified in drought trees with and without bacteria showed clusters that partially overlapped (*Figure 3—source data 2* and *Figure 3—figure supplement 1A*). When samples from irrigated trees were also considered, they behaved similarly. Of the semi-polar compounds that changed significantly under drought, 20 were further identified (*Figure 3—figure supplement 1*). Finally, three semi-polar metabolites

**Table 1.** Dynamic changes in selected metabolites across treatments (DB, drought and bacterial inoculation; D, drought; I, irrigation) and along the experiment (drought and recovery following re-irrigation).

t-Test results are for pairwise comparisons between the relative changes of a metabolite between two treatments. Significant values are in boldface. Log 2 fold changes of the metabolites are for pairwise comparisons between two treatments.

| Metabolite | t-Test | | Log 2 fold change | | | |
|---|---|---|---|---|---|---|
| | **Drought DB/D** | **Drought D/I** | **Drought DB/D** | **Drought D/I** | **Re-irrigation DB/D** | **Re-irrigation D/I** |
| 2,3-Dihydroxybenzoate | **0.013** | 0.759 | 0.12 | 0.01 | –0.02 | –0.07 |
| 3,4-Dihydroxymandelate | 0.675 | **0.010** | –0.04 | 0.19 | 0.07 | 0.14 |
| 3,5-Dimethoxyphenol | **0.048** | **0.040** | –0.06 | 0.05 | 0.01 | 0.07 |
| Citrulline | 0.065 | **0.005** | 0.33 | –0.44 | –0.10 | 0.19 |
| *N*-acetylserine | **0.043** | **0.029** | 0.14 | –0.13 | –0.02 | 0.06 |
| D-chiro-inositol | **0.040** | **0.008** | 0.23 | –0.16 | –0.05 | 0.09 |
| Quinate | **0.025** | **0.014** | 0.26 | –0.25 | –0.08 | –0.14 |
| Nicotinate | **0.039** | **0.011** | 0.05 | –0.03 | 0.02 | 0.02 |
| *p*-Coumaric acid | 0.227 | **0.001** | 0.13 | –0.21 | –0.04 | 0.06 |

increased significantly in inoculated drought trees compared to drought trees without bacteria: isomaltose, clerodane diterpene, and protocatechuic acid (dihydroxybenzoic acid) (compounds 14, 17, and 20, respectively, in *Figure 3—figure supplement 1C*).

## Metabolites exuded by tree roots promote growth of rhizosphere bacteria

It was of interest to examine for potential utilization of exudates by *P. stutzeri* and *B. subtilis*. Among the 44 drought-specific root exudate metabolites identified, nine secondary metabolites that had high abundance relative to the baseline measurement were selected. These metabolites represent diverse chemical classes and different levels across the treatments and stages of the experiments (*Figure 3B*, *Figure 3—source data 1*; *Table 1*). Eight of the metabolites showed significant differences between irrigated and drought trees (typically decreasing with drought, except for 3,4 dihydroxymandelate and 3,5-dimethoxyphenol, which increased; *Figure 3B*, *Figure 3—source data 1*; *Table 1*). A single metabolite, 2,3-dihydroxybenzoate, did not change with drought, however increased with bacterial inoculation under drought. Of the other eight metabolites, six also increased with bacterial inoculation under drought, although not significantly in citrulline and *p*-coumaric acid. Again, 3,4 dihydroxymandelate and 3,5-dimethoxyphenol differed by decreasing with bacteria, significantly for the latter. Following re-irrigation, none of the metabolites changed significantly between treatments, however citrulline and 3,4 dihydroxymandelate increased under re-irrigation (*Figure 3C*; *Table 1*, t-test results not shown).

Bacterial cultures were grown on defined media, or, alternatively, deprived from either glycerol or glutamate as carbon or nitrogen source, respectively, which were substituted by root exudation compounds. Among the nine metabolites, the three phenolic acid compounds (2,3-dihydroxybenzoate; 3,4 dihydroxymandelate [protocatechuic acid], and 3,5-dimethoxyphenol), the two amino acid derivatives (*N*-acetylserine, L-citrulline), and the shikimate pathway acid (D-Quinic acid) were found to support growth of *P. stutzeri* (*Figure 4*). Growth of *B. subtilis* cultures in the media with 3,4-dihydroxymandelate, with or without 2,3-dihydroxybenzoate, was significantly higher compared to media containing glycerol as a carbon source (2.5- to 3-fold increase; p<0.01). Using 2,3- dihydroxybenzoate alone was less efficient than glycerol for *B. subtilis* growth, and similar for *P. stutzeri* (*Figure 4A*). Next, the metabolites *N*-acetylserine, L-citrulline, and D-quinic acid yielded similar growth rate compared to glutamate as nitrogen source, and more so for *P. stutzeri*. In *B. subtilis* cultures, there was a delay in growth with L-citrulline, and a later decrease. D-quinic acid caused a delay in growth initially but eventually reached higher growth than the control (*Figure 4B*). *p*-Coumaric acid was used as sub-optimal carbon source for *P. stutzeri*, whereas nicotinate and D-chiro-inositol did not serve as carbon or nitrogen source (*Figure 4—figure supplement 1*). Nevertheless, these two compounds might be still involved in other manners in bacterial life around the roots.

## Leaf, root, and soil elements respond to drought and bacterial inoculation

Levels of P, Fe, and Zn were significantly lower in leaves in drought compared to irrigated trees during the middle of the drought period (27.05). This effect was mitigated in inoculated trees under drought for P and Fe, but not for Zn (*Figure 5A*, *Figure 5—source data 1*). For Mn, the drought effect was not significant, and the bacterial inoculation rather slightly decreased its leaf content. Key elements were further visualized in leaf and root samples. Elemental distribution images (heat maps) of *Cupressus* leaves showed Mn concentrations in leaf meristems and Fe spreading across the leaves as particles, with leaves of irrigated trees showing higher K levels (*Figure 6A and B*). Elemental maps of *Cupressus* roots under irrigation showed co-localization of Fe, K, and Mn (*Figure 6C*). In *Cupressus* roots under drought, Fe and Mn were localized to outer portions of the root, whereas K and Cu were mostly at the inner regions (*Figure 6D*). Overall, roots of irrigated trees had higher Fe, Mn, and Cl than roots of drought trees, with no change in Zn (*Figure 6D*).

The bioavailability of phosphate was measured by assaying the Olsen-P concentration in the soil. It was found to be responsive to drought, decreasing by twofold compared to irrigated soil (p<0.05; *Figure 5B*). Importantly, Olsen-P of dry soil increased fourfold when inoculated with soil bacteria (p<0.01). Re-irrigation increased Olsen-P back to its level in wet soil and there was no effect of adding bacteria at this stage. There was a positive correlation between leaf and soil P around drought trees

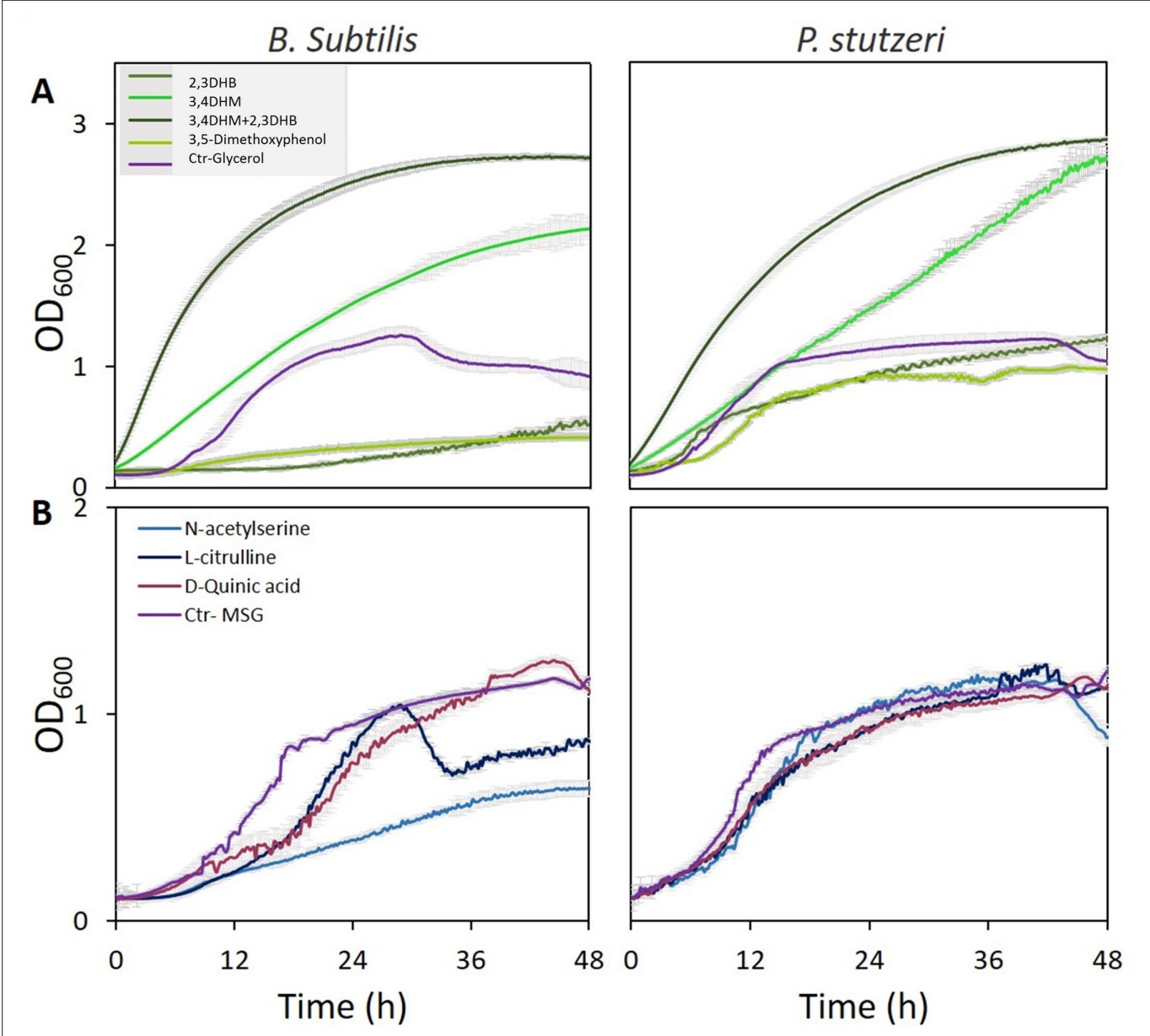

**Figure 4.** Specific metabolites in tree root exudates enhance bacterial growth. Growth curves of *Bacillus subtilis* (left) and *Pseudomonas stutzeri* (right) in defined media with specific metabolites identified in root exudate blends of drought-exposed *Cupressus sempervirens* saplings as exclusive carbon (**A**) or nitrogen (**B**) source. Glycerol and monosodium glutamate (MSG) were used as controls for carbon and nitrogen sources, respectively. Root-derived metabolites are 2,3-dihydroxybenzoate (2,3 DHB); 3,4 dihydroxymandelate (3,4 DHM); their 1:1 mixture; 3,5-dimethoxyphenol; *N*-acetylserine; L-citrulline; and D-quinate. Concentrations are 0.4% by weight for all compounds. OD$_{600}$, optical density at a wavelength of 600 nm (n=4).

The online version of this article includes the following figure supplement(s) for figure 4:

**Figure supplement 1.** Growth curves of *Bacillus subtilis* (left) and *Pseudomonas stutzeri* (right) in defined media with specific metabolites identified in root exudate blends of drought-exposed *Cupressus sempervirens* saplings as exclusive carbon (**A**) or nitrogen (**B**) source.

under re-irrigation (r=0.58 and 0.67 for inoculated and un-inoculated trees, respectively). In irrigated trees, r values ranged between 0.03 and 0.23. The correlations between leaf P and photosynthesis were typically <0.50, and increased following the first bacterial inoculation (r=0.70 and 0.65 for inoculated drought and irrigated trees, respectively). In later time points, photosynthesis correlated well with leaf P only in drought trees without inoculation (r=0.80–0.88).

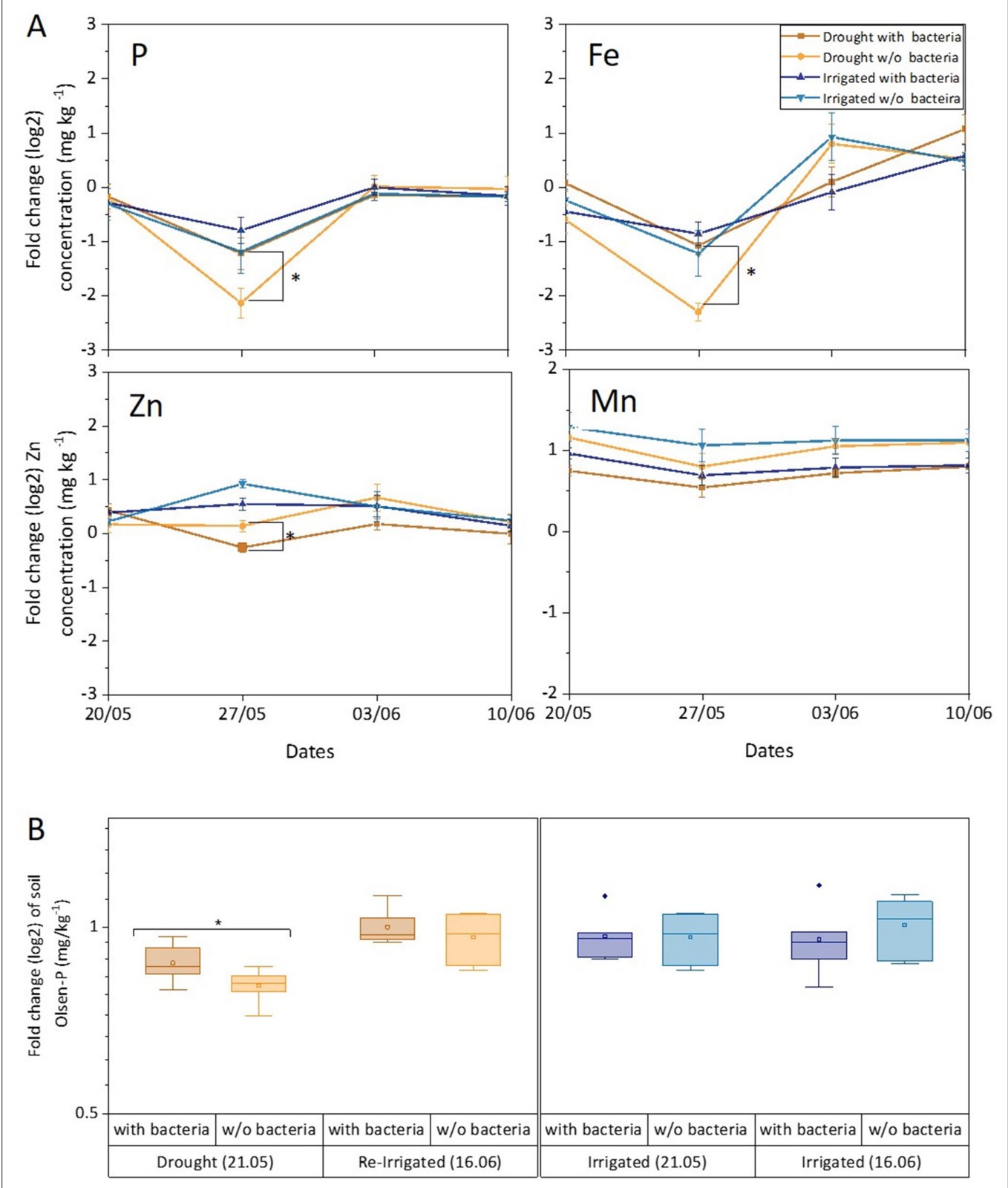

**Figure 5.** Leaf and soil elements respond to drought and bacterial inoculation. Leaf elements (**A**) and soil phosphorous (**B**) of irrigated (blue shades) and drought-exposed and re-irrigated (brown shades) *Cupressus sempervirens* saplings, with and without bacterial inoculations. Values are log 2 of fold change from the baseline leaf elements and soil phosphorous (at the beginning of the experiment) in mg kg⁻¹ concentrations. Asterisks indicate

*Figure 5 continued on next page*

*Figure 5 continued*

significant differences based on by two-way ANOVA performed with Tukey's HSD test or Bonferroni test (n=6, p<0.05). See also **Figure 5—source data 1**.

The online version of this article includes the following source data for figure 5:

**Source data 1.** Statistical analysis of leaf elements.

## Discussion

Soil drought was imposed on young *Cupressus* trees growing outside in forest soil, decreasing their soil water and P availability (**Figure 1—figure supplement 2**, **Figure 5B**). Trees responded with strong reductions in hydraulic and photosynthetic activity, as well as in leaf nutrients (**Figure 2—figure supplement 1**, **Figure 5A**). Despite the lower carbon uptake, trees continued to exude altered metabolite blends (**Figure 2**, **Figure 3**). Soil bacteria associated with tree roots and grew preferentially in the rhizosphere (than in the bulk soil; **Figure 1**), where their presence transiently increased with root exudates (**Figure 2**, **Figure 7**). Among ~150 specific metabolites, many decreased in response to drought, but increased back when trees were inoculated with bacteria (**Figure 3**). Out of nine of these metabolites that were tested in vitro on soil bacteria, six were used by bacteria as either carbon or nitrogen source, sometimes enhancing growth more than the standard source (**Figure 4**). In turn, drought trees that were inoculated had increased biomass increment, as well as leaf nutrients and slightly higher photosynthetic activity before re-irrigation (**Figure 5A**, **Figure 2—figure supplement 1**). Upon re-irrigation, trees gradually recovered, while drastically decreasing their root exudates, both in quantity and quality. While our manipulation changed the native microbial community, it permitted the direct observation of the root-bacteria interplay. This interplay was over a native background, rather than over a sterilized or synthetic media, to simulate the natural conditions as much as possible. Our approach introduces more disturbance than, for example, a microbiome approach, however the latter would not permit direct observation of specific strains and their interactions with tree roots, as reported here.

### Costs and benefits for trees and soil bacteria

Our results confirm our hypothesis that desiccated trees will suffer drought stress both belowground (reduced water and nutrient uptake; **Figures 5 and 6**) and aboveground (closed stomata and reduced carbon and water fluxes; **Figure 2—figure supplement 1**). The hypothesis that bacterial communities will attach to the tree roots (**Figure 1**) and remain there over time was also confirmed, albeit bacterial abundance decreased with time after inoculation (**Figure 1**; see Study limitations below). However, this decrease is expected, considering that inoculations contained bacteria in numbers exceeding the system's capacity, as previously shown (**Bever et al., 2012**). Next, root exudation increased in response to bacterial inoculations (**Figure 2**). The phenomenon of stimulation of exudates by bacteria was demonstrated for exudation of primary metabolites (**Canarini et al., 2019**). However, the level of exudation was lower under drought, in contrast to our former field observations (**Jakoby et al., 2020**). It is likely that the imposed drought in our experiment was harsher than the dry season conditions in the field, where roots can explore deeper soil layers. Still, the sharp decrease in exudation rate during re-irrigation highlights the relatively higher exudation rate under the combination of drought and bacterial inoculation. Importantly, when calculating the exudation rate as percent of carbon uptake, the rates of inoculated drought trees were actually threefold higher compared to irrigated trees. Root exudates included carbohydrates and organic acids that fed bacterial communities (**Figures 3 and 4**). Yet, unexpectedly, most root exudates under drought were secondary, rather than primary metabolites. However, this is in agreement with a similar shift observed in the exudate metabolomes of drought-exposed oak trees (**Gargallo-Garriga et al., 2018**). Indeed, phenolic acid compounds and amino acid derivatives proved to be superior carbon and nitrogen sources than a sugar and an amino acid.

Finally, desiccated trees supplemented with soil bacteria showed better nutrition in P and Fe (**Figure 5**); but, unexpectedly, did not recover faster than without bacteria. Thus, photosynthesis recovered faster in drought trees that were not inoculated than those that were (**Figure 2—figure supplement 1**). Considering that inoculated trees invested significantly more carbon into the rhizosphere than un-inoculated drought trees, it is possible that this carbon cost came at the expense

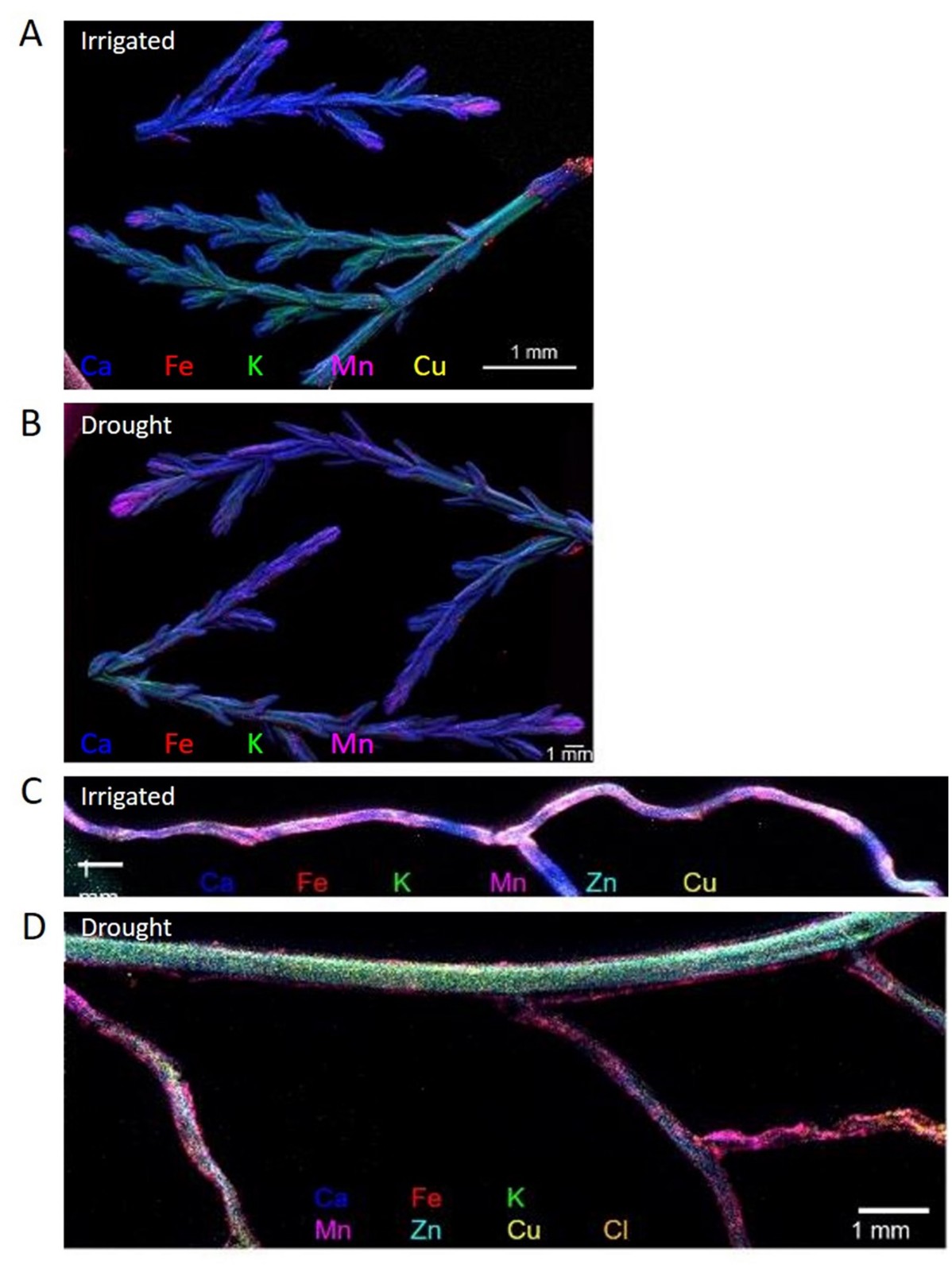

**Figure 6.** Composite elemental distribution images of irrigated and drought-exposed leaves (**A, B**) and fine roots (**C, D**) of *Cupressus sempervirens* saplings. Elemental distribution images (heat maps; a single color per element) were created at 20 µm resolution and scan area of 500 µm with a copper target to detect Fe, Mn, Ca, K, and Cl, and molybdenum or gold target to detect Cu and Zn. The mapping was performed using an X-ray fluorescence microscope applying an ultra-high brightness X-ray source with multiple X-ray targets.

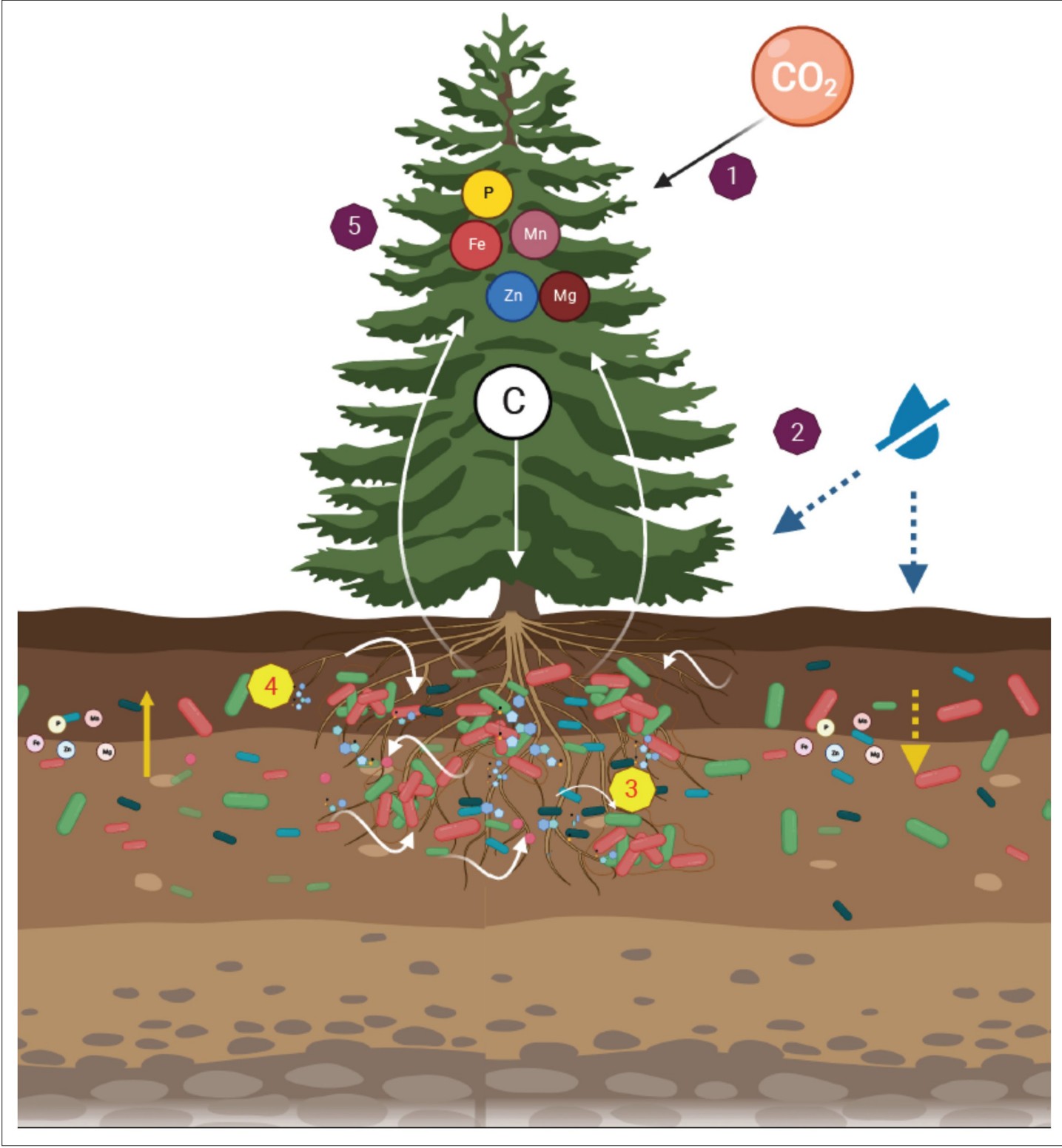

**Figure 7.** A dynamic rhizosphere interplay between tree roots and soil bacteria under drought stress. Leaf $CO_2$ assimilation distributed across the tree tissues (1). Under soil drought, leaf gas exchange and tree growth were inhibited (2). Trees recruited root-associated bacteria through changes in exudation rates and composition (3). The metabolites exuded by roots promoted growth of rhizosphere bacteria (4). In turn, the drought-induced reduction in leaf mineral concentrations (e.g. phosphorous and iron) were mitigated by bacteria (5).

The online version of this article includes the following source data for figure 7:

**Source data 1.** Statistical analysis of leaf assimilation, root exudation, and their relationships.

of internal tree reserves, later creating a 'recovery penalty' for the trees (*Gessler et al., 2020*). The carbon cost of root exudation has already been accounted for in relation to P foraging (*Dijkstra et al., 2013*; *Wang and Lambers, 2020*), where rhizodeposition may be used for P scavenging rather than for decomposition of soil organic matter. On the other hand, slower recovery might indicate a more controlled acclimation response (*Bastida et al., 2019*; *Gessler et al., 2020*). Overall, the rhizosphere interplay between trees and bacteria improved tree nutrition under stress, however the physiological benefit was more subtle than rigorous, mostly in better sustaining photosynthesis under drought, as shown before (*Morgan et al., 2005*; *Boutasknit et al., 2020*). Correlations between leaf P and photosynthesis indicated that the improved nutrition was associated with improved physiology, especially following the inoculation under drought, and later in drought trees without inoculation, where individuals with higher leaf P had higher photosynthesis. On the bacterial side, benefits include a growth habitat (*Figure 1A*), as well as food source (*Figure 5*), but costs are harder to pinpoint. By congregating in the rhizosphere, and specifically, on the roots, bacteria should face higher competition for resources. High bacterial density was shown to increase competition for iron in soil (*Gu et al., 2020*).

## A dynamic rhizosphere interplay between tree roots and soil bacteria

Four lines of evidence in our study support a mutual interaction between trees and bacteria: (1) bacteria were physically attached to root surfaces; (2) bacteria were always ~10-fold more abundant in the rhizosphere than the bulk soil; (3) root exudation increased following inoculation, and was higher during drought than following it (*Figure 1—source data 1*, *Figure 7—source data 1*); and (4) many metabolites were enriched following inoculation in drought trees. The higher root exudation during drought than under re-irrigation is not trivial, considering that wet soil provides a better environment than dry soil for both tree roots and bacteria development (*Gao et al., 2021*). Our temporal resolution and complex experimental setup do not address the question if exudates attracted the bacteria, or, alternatively, whether the bacteria induced root exudation. It is likely that both processes occurred simultaneously. One must also consider alternative explanations, for example, that exudates act directly on soil elements, rather than solicited by bacteria. During drought, changes in soil water content could become either beneficial (increased Mn and P availability) or harmful (decreased Zn availability) to plant nutrition (*Misra, 1999*). Tree roots exude a plethora of secondary metabolites into the rhizosphere, which aid in the mobilization and uptake of essential macro-elements as N, P, and microelements like Mg, Mn, Zn, and Fe (*Michalet et al., 2013*). Several studies reported that beneficial rhizosphere bacteria drive an accumulation of elements (*Philippot et al., 2013*; *Igiehon and Babalola, 2018*). Although we showed that multiple exudate metabolites promoted bacterial growth, this does not mean that all of these metabolites act as specific cues for specific bacterial strains in the rhizosphere. However, the intimate attachment of bacteria to roots shown here should ensure the harvest of these metabolites by bacteria.

Plant-microbe interactions at the root level exert strong effects on the whole plant and on nutrient cycling, which are mostly explained by root traits (*Teste et al., 2017*; *Guyonnet et al., 2018*). Here, we greatly enhanced the abundance of two bacterial strains, each showing unique dynamics. Previous works found that during the dry season, root-associated communities show elevated abundance for *Bacillus*, whereas in the rainy season, Proteobacteria increase (*Shakya et al., 2013*; *Taketani et al., 2017*). In our experiment, *P. stutzeri* decreased under drought faster than *B. subtilis*, but took better advantage of the root exudate metabolites 3,5-dimethoxyphenol, L-citrulline, *N*-acetylserine, D-quinic acid, and *p*-coumaric acid (section below). The rhizosphere roles of these two strains might be complementary: while *Pseudomonas* is abundant under humid conditions (*Mendes et al., 2013*; *Xu et al., 2018*), *Bacillus* dominates plant microbiomes under arid conditions, where *Pseudomonas* cannot survive.

## The chemistry of tree-microbe interactions in the rhizosphere

Interestingly, the phenolic acid compounds that promoted bacterial growth the most (2,3-dihydroxybenzoate; 3,4-dihydroxymandelate; and 3,5-dimethoxyphenol) did not decrease with drought while others did. This observation supports the view of their role in feeding the root bacteria as part of a tree drought resistance strategy. 2,3-Dihydroxybenzoate is a bacterial growth factor (*Young et al., 1967*). It is the biosynthetic precursor of petrobactin, an unusual 3,4-dihydroxybenzoate catecholate siderophore essential for growth of *Bacillus anthracis* and *Pseudomonas putida* (*Garner*

*et al., 2004*; *Neal et al., 2012*). Hydroxybenzoic acids also are known as being allelopathic and anti-fungal compounds of plant exudates (*Yu and Matsui, 1994*; *Mandal et al., 2009a*). They have been characterized as defensive secondary metabolites that are released by roots of cereals such as wheat and maize to alter root-associated fungal and bacterial communities (*Neal et al., 2012*; *Hu et al., 2018*). Benzoic acid can stimulate growth of a *Pseudomonas* species (*Kamilova et al., 2006*), and *p*-hydroxybenzoate can serve as a sole carbon source for the growth of another *Pseudomonas* species (*Harwood et al., 1984*). Recently, *Ankati and Podile, 2019*, showed that colonization of groundnut roots by related *Bacillus* and *Pseudomonas* strains resulted in exudation of metabolites such as benzoic and salicylic acids (both identified here) that facilitated root colonization, suppressed fungal growth, and promoted plant growth. Other metabolites identified such as quinate and citrulline were either among the most abundant in exudate blends throughout the experiment, or those that increased in inoculated trees. Quinate is a major compound in *Pinus radiata* exudates and was shown to stimulate over 700 soil bacterial community taxa (*Shi et al., 2011*; *Zhalnina et al., 2018*) and citrulline was identified in soybean exudates under P deficiency (*Tawaraya et al., 2014*).

Bacterial metabolism plays a key role in root colonization, and carbohydrate metabolism genes of *Pseudomonas simiae* were linked to its colonization of *Arabidopsis* roots (*Naylor and Coleman-Derr, 2017*). Those findings are consistent with our finding of viable carbon and nitrogen sources supporting bacterial growth among root exudates. However, other metabolites were not supportive of growth of both strains. Recently, *Ma et al., 2018*, demonstrated the effect of nicotine from tobacco root exudates on an antagonistic bacterium for its root colonization, and control of chemotaxis of soil-borne pathogens. *p*-Coumaric acid has been identified in plant root exudates or residues and in soils around many plant species (*Hao et al., 2010*). 4-Coumaric acid was identified with anti-fungal activity in tomato roots (*Mandal et al., 2009b*). *p*-Coumaric acid increased the relative abundances of microbial taxa with phenol-degrading capability (*Zhou et al., 2017*). Considering our observations of inhibition of *Bacillus* growth by *p*-Coumaric acid, it might be engaged in tree root interaction through generating negative tree-soil microbial interactions. D-chiro-inositol is one of multiple phospho-inositides, essential metabolites, as well as labile messengers that regulate cellular physiology, and are involved in tolerance to abiotic stress (*Stevenson et al., 2000*).

Among the many exudate compounds elucidated in our analysis, metabolites other than the nine tested here were of interest. These include indole-3-acetic acid (IAA), namely the plant hormone auxin, which is known as active in hydrotropism. Here, IAA had a moderate level which decreased both under drought and re-irrigation. Auxin can be produced in the rhizosphere both by roots and PGPRs (*Tsunoda and van Dam, 2017*). The amino acid proline, a known plant osmoprotectant under drought, had a moderate level, which decreased under drought, mildly recovering under re-irrigation both with and without inoculation. Another notable metabolite is shikimate, whose pathway includes the aforementioned quinate, and has already been shown to increase root colonization by beneficial *Bacillus amyloliquefaciens* (*Rolfe et al., 2019*). Here, shikimate had a high level, mildly decreasing under drought. Gamma aminobutyric acid is another well-known metabolite, which was identified here at a high level, decreasing under irrigation and inoculation. Last, we identified trigonelline, which is known to induce nod gene in *Rhizobium* bacteria.

## Study limitations

We offered a glimpse into the rhizosphere, which was not, however, free of limitations. First, our approach ignored the native rhizosphere microbiome, which probably affected many of the studied parameters. In a complementary fungal study on these young trees, we were unable to identify mycorrhizal partners, however the root pathogen *Plectosphaerella cucumerina* was identified. Nevertheless, the two bacterial species that we used were typical root colonizers of *Cupressus*, and the *Pseudomonas* strain was isolated from the very same forest soil. Second, manipulating the rhizosphere with inoculations can also be criticized as creating disturbance. Moreover, except for the re-inoculated irrigated trees, bacterial communities were not stable. Still, the two bacteria species were released into the forest soil to compete as any other microbial species in the soil system. Third, our sample size was relatively small, with six trees per treatment, however compensated by the Rhizobox design and the intensity of measurements. Fourth, passive exudation and leakage from drought-stressed roots cannot be ruled out in our experiment. However, the observed shift in metabolites and the sensitivity to inoculations support an active exudation. Fifth, whether our results are transferable from saplings

to mature trees is questionable. We argue, however, that studying saplings is equally important, as establishment is often the bottleneck for tree survival in the forest (*Pozner et al., 2022*), and it is the period where microbial partners make the most difference. This is increasingly important in the case of shallow-rooted tree species such as *C. sempervirens* (*Rog et al., 2021*). In the forest, mycorrhizal fungi, which were missing here, can have large effects too (*Meier et al., 2013*). Specifically, *C. sempervirens* hosts a diversity of active arbuscular mycorrhizal species (*Avital et al., 2022*). While the latter generally help in tree P nutrition, their activity is mostly reduced under drought. Future studies should test the functions of the three-kingdom interactions among trees, fungi, and bacteria. *Finally, our exudate collection technique cannot entirely rule out metabolites from bacterial, rather than plant, origin, nor can our incubations account for loss of exudates to bacterial consumption. Root-free and dissected root controls significantly increase our confidence in the results, but both false positives and false negatives are possible.*

## Conclusion

In this multidisciplinary study we combined advanced approaches in microbiology (genetic modifications and epifluorescence), plant physiology (leaf gas exchange and exudate collection), and organic chemistry (metabolic profiling) to decipher the dynamic rhizosphere interplay between tree roots and soil bacteria under drought stress. The novelty of our results is by providing a holistic view of the consequence of events in the rhizosphere, from tree water and nutritional shortages, through root exudation of specific metabolites (e.g. phenolic acid compounds) and their benefit to bacterial growth, and finally, to improved tree nutrition (leaf P and Fe). Our next steps aim at further elucidating the mechanisms utilized by both trees and bacteria for their mutual communication and cooperation in the rhizosphere.

## Materials and methods
### Experimental design and plant material

The study was performed in custom-made 25 L containers termed 'Rhizoboxes' in a net-house in the Weizmann Institute of Science, Rehovot (Israel). These 25 cm long × 25 cm wide × 40 cm tall containers were produced ad hoc from transparent poly(methyl 2-methylpropenoate) and were filled with a mixture (1:1) of washed siliceous sand and forest soil, to preserve the naturally occurring fungi and bacteria in the media, while preventing soil clogging. For every 10 L of mixed soil, 0.5 L Tuff was added to improve aeration. Soil properties were measured before and in the end of the experiment (*Table 2*). The forest soil was collected from Harel forest, located at the Judean foothills (31° 43′N 34° 57′E, 320 m elevation), 4 km south-west of Beit Shemesh, Israel. Rhizobox containers were covered in black plastic bags, to facilitate root growth, uncovered only during root sampling or exudate measurements. The experiment was conducted during May-July 2019. Weather conditions in Rehovot during May-July 2019 were the following: overall minimum, maximum, and average temperatures were 11.3°C, 34.5°C, and 25°C, respectively, average VPD was 2.00 kPa, and no rain (Beit-Dagan

**Table 2.** Soil properties from pots where saplings were grown under treatments of drought or irrigated with or without bacteria.

**(A) Soil chemistry: pH, electrical conductivity (EC), sodium absorption ratio (SAR), and minerals content (Cl, Na, Ca, Mg, N-NH4, N-NO₃, K concentrations)**

| Soil | Treatment | pH | EC (ds m⁻¹) | Cl (mg kg⁻¹) | Na (g L⁻¹) | Ca (mg kg⁻¹) | Mg (mg kg⁻¹) | SAR (mg kg⁻¹) | N-NH₄ (mg kg⁻¹) | N-NO₃ (mg kg⁻¹) | K of CaCl₂ (mg kg⁻¹) |
|---|---|---|---|---|---|---|---|---|---|---|---|
| Pre (n=3) | Pre (n=3) | 7.5±0.06 | 1.88±0.07 | 87.7±3.0 | 54.4±3.6 | 750.7±8.8 | 11.8±0.3 | 2.8±0.09 | 18.9±3.0 | 25.63±3.4 | 49.43±3.9 |
| Drought | With bacteria | 7.5 | 2.51 | 170.0 | 179.4 | 1209.9 | 17.2 | 7.2 | 23.8 | 3.7 | 39.6 |
| Drought | W/o bacteria | 7.7 | 2.64 | 188.9 | 80.7 | 1222.4 | 16.9 | 3.2 | 19.1 | 3.5 | 37.3 |
| Irrigated | With bacteria | 7.6 | 6.31 | 513.6 | 54.4 | 3358.5 | 49.2 | 1.3 | 37.2 | 2.5 | 42 |
| Irrigated | W/o bacteria | 7.5 | 6.27 | 544.7 | 191.1 | 2716.3 | 51.1 | 5.1 | 41.9 | 3.6 | 39.6 |

**(B) Physical structure: sand, silt, and clay content, calcite content (CaCO₃) and soil porosity (SP).**

| Soil | Treatment | Sand % | Salt % | Clay % | CaCO₂ | SP |
|---|---|---|---|---|---|---|
| Pre (n=3) | Pre (n=3) | 9±3.46 | 19.67±2.08 | 71.33±5.51 | 3.67±1.15 | 37.33±2.08 |

meteorological station, 10 km north of the research institute; *Figure 1—figure supplement 1*). Global solar radiation was 300–350 W m$^{-2}$, reduced by ~15% by the net-house.

Twenty-four 2-year-old *C. sempervirens* saplings were obtained from the Jewish National Fund, Israel Forest Service (KKL) nursery and transplanted into Rhizoboxes in the Weizmann Institute of Science campus (Rehovot, Israel) in September 2018. Prior to that, seedlings were germinated from seeds collected at Beit She'arim in the Galilee, Israel, and were transferred to plastic 'quick-pots 585' (200 mL plugs, 5 cm×5 cm) in the KKL nursery. There, the plants were grown with 2% starter fertilization and irrigated with fertilization until September 2018. The saplings were kept at the net-house, one sapling per Rhizobox, under optimal irrigation of 1.2 L day$^{-1}$ for a 7-month period, to allow root growth into the mixed forest soil. Saplings were 10.3±0.6 mm in stem diameter and ~50–100 cm in height. Next, saplings were equally divided into four groups and were placed randomly in the net-house. Two groups (12 trees) were exposed to drought conditions, induced by withholding irrigation for 32 days. The volumetric soil water content was measured on each measurement day with moisture sensor (EC-10, Decagon Devices, Pullman, WA) (*Figure 1—figure supplement 2*). The timing of re-irrigation was determined according to the trees' physiological conditions, immediately after transpiration and stomatal conductance decreased to zero (*Figure 2—figure supplement 1*). The other two groups (12 trees) were irrigated continuously with 1.2 L day$^{-1}$. On the 9th day from the irrigation cessation (May 20, 2019), one drought group and one irrigated group were inoculated with a 1 L bacterial solution (see below). The same groups were inoculated again on the 5th day after re-irrigation (June 16, 2019). Inoculations were made with two native soil bacteria strains over a forest soil background, to simulate natural conditions while permitting quantification and visualization of the bacteria and their interactions with tree roots. Although reducing the complexity of the community, it was shown that pairwise interactions between soil bacteria predict well the community structure (*Friedman et al., 2017*). All measurements were conducted in the following days: (1) 11 days before irrigation cessation (day –11, 2 May); (2) 24 hr after the first inoculation (day 10, 21 May); (3–5) once a week following irrigation cessation (days 16, 23, and 30, on 27 May, 3 June, and 10 June, respectively); (6) 4 days after re-irrigation (day 36, 16 June); (7) following the second inoculation (day 37, 17 June), and (8) 26 days after re-irrigation (day 58, 8 July). Bacteria counts were made 1, 3, 7, and 15 days (days 10, 13, 16, and 24), following the first inoculation, and similarly following the second inoculation (days 38, 41, 44, and 51). Root exudates were collected at 5 days before drought treatment (day –5, 7 May), 1 day after the first inoculation (day 10, 21 May), and 4 days after re-irrigation (day 36, 16 June; *Figure 1—figure supplement 2*). Total biomass assessment was determined on six saplings, which were sacrificed before the experiment started and at the end of the experiment, on all saplings. The saplings were separated from the soil and dried at 60°C for 3 days in order to measure biomass.

## Soil water content and soil properties

Soil water content (%, v/v) was measured using a dielectric constant EC-5 soil moisture sensor connected to an Em50 Data logger (Decagon Devices Inc, Pullman, WA) which was programed to record observations at 10 min intervals. The sensors were located in the Rhizobox at 15 cm depths with two repeats for each treatment. Prior to the experiment, soil samples were collected from three different locations of the soil mixture pile, before Rhizoboxes were filled, and sent for soil analysis at Gilat Field Services Laboratory, Israel. Soil structure was 9/20/71% sand/silt/clay, with 3.7% CaCO$_3$ and 37% porosity. Electric conductivity was 1.9 dS m$^{-1}$; pH was 7.5; sodium absorption ratio 2.1; N-NO$_3$, N-NH$_4$, K, and Olsen-P content was 18.9, 25.6, 49.4, and 11.7 mg kg$^{-1}$, respectively. Cl, Na, Ca, and Mg content was 224.8, 6.1, 270.1, and 30.4 mg L$^{-1}$, respectively.

## Soil phosphorous extraction and quantification

Soil samples were collected from each container at three time points during the experiment. The soil was oven dried (60°C for 48 hr) and ground to pass a 2 mm sieve. Each sample was divided into three replicates of 1 g. Bicarbonate (30 mL of 0.5 M solution) was applied and mixed for 1 hr, for extraction according to the Olsen method (*Olsen et al., 1954*). The extraction was centrifuged at 10,000 *g* for 10 min, and then an aliquot was transferred to a new tube, and pH was adjusted to 6.0 by H$_2$SO$_4$. After extraction, total phosphate was measured using the Abcam colorimetric phosphate assay kit (ab65622, detection limit of 1 μM), and quantified with a Tecan Infinite M200 plate reader (Tecan, Grödig, Austria).

## Live imaging of bacterial root colonization using confocal microscopy

Bacterial root colonization was visualized and photographed by a confocal super-resolution (CLSM) Nikon A1 + laser system equipped with Plan-Apochromat ×10/NA0.45 (Nikon, Tokyo, Japan). *B. subtilis* cells expressing GFP were irradiated using a 488 nm laser beam, while *P. stutzeri* cells expressing mCherry were irradiated using a 555 nm laser beam. For each experiment, both transmitted and reflected light were collected. System control and image processing were carried out using NIS-Elements C software version 4.0 (Nikon).

## *Pseudomonas* strain isolation, characterization, and construction

Samples of forest soil were collected from Harel forest (5–15 cm depth) in order to isolate native PGPR *Pseudomonas* strains from tree roots. Soil aliquots of 10 g each were placed in 50 mL PBS (phosphate-buffered saline) and mixed vigorously for 1 min. Subsequently, serial dilutions were plated on LB agar plates and were incubated at 23°C for 24 hr. *Pseudomonas* strains were isolated and genomic DNA was purified using the Wizard Genomic DNA Purification Kit (Promega, Madison, WI) according to the manufacturer's instructions. For DNA sequencing, the 16S rRNA genes were amplified using PCR with the forward primer (5-799F AACMGGATTAGATACCCKG) and the reverse primer (6-1192R ACGTCATCCCCACCTTCC') (*Chelius and Triplett, 2001*). Sequences were aligned with other sequences downloaded from the GenBank database using NCBI BLAST. *P. stutzeri* was transformed with plasmids pTns2 and pUC18T-mini-Tn7T-Gm-mCherry, to generate *P. stutzeri* (*attTn7::mCherry*) strain with antibiotic resistance as selectable marker, and fluorescence for visual detection (*McFarland et al., 2015*).

## Bacterial strains, inoculum preparation, and bacterial quantification

*B. subtilis* and *P. stutzeri* were used in this study to inoculate the trees. These strains represent two major groups of soil bacteria, that is, the phylum Firmicutes, and the class Gammaproteobacteria, respectively. In addition, both strains offer the advantage of cultivation and genetic manipulation, and both are known to interact with roots of other plants (*Weller et al., 2002*; *Kravchenko et al., 2003*; *Rudrappa et al., 2008*; *Zhang et al., 2014*; *Sasse et al., 2017*; *Zhang et al., 2017*; *Mhlongo et al., 2018*). *B. subtilis* (NCIB 3610) containing constitutive GFP (*amyE::Phyper-spank-gfp-cm*) was a gift from R Losick (Harvard Medical School, Boston, MA) and *P. stutzeri* was isolated from soil (above). Although the specific *B. subtilis* strain used here was not recovered from the forest soil, we did identify another *B. subtilis* strain in our forest soil. In multiple screens of rhizosphere bacteria in our forest site, a total of 54 bacterial strains were identified on plates. Of which, *B. subtilis* and *P. stutzeri* were the most consistent, evidencing their important role in our system. Inoculum was prepared by growing the bacterial isolates *B. subtilis* and *P. stutzeri* in LB broth at 37°C for 16 hr, followed by centrifugation (4000 rpm for 10 min at 23°C), washing and re-suspension in carbon-free nutrient solution to obtain a cell density of $5 \times 10^8$ colony forming units (CFU) mL$^{-1}$ (OD$_{600}$ = 1). Equal amounts of bacterial cultures of the two species were mixed and were spread homogeneously in the Rhizobox. To avoid any bias, trees which were not inoculated received the same amount of water as in the inoculation solution at the time of inoculation. To isolate the two bacterial species, soil and roots (10 g of each) were collected into 10 mL PBA and homogenized by 1 min vortex followed by 10 min sonication. The homogenate was serially diluted in sterile PBS and the dilutions were plated on two LB agar medium supplemented with gentamicin (Gm, 30 µg mL$^{-1}$) to isolate *P. stutzeri* or chloramphenicol (Cm, 25 µg mL$^{-1}$) to isolate *B. subtilis*. Total bacterial counts obtained were expressed as log CFU g$^{-1}$ of dry weight of soil or root. The relative abundance was estimated by counting each strain of bacteria separately and diving by the total amount of bacteria.

   To construct growth curves in response to specific plant metabolites identified in root bacterial exudates, the strains were grown in MSgg medium (*Bloom-Ackermann et al., 2016*) containing MS salts and 125 µM FeCl$_3$, either with glycerol (0.4% w/v) and glutamate (0.4% w/v) or with root tree metabolite as carbon or nitrogen source (0.4% w/v). Metabolites were identical with those identified in root exudates (*Table 1*), except for nicotinate, which was tested in the form of nicotine. The cultures were inoculated at an initial OD$_{600}$ = 0.04 and then incubated at 25°C with shaking at 150 rpm for up to 48 hr in the Tecan plate reader. OD$_{600}$ were measured continuously over 48 hr.

## Leaf gas exchange and water potential

Leaf physiology was monitored once a week for all the trees in the experiment by leaf gas exchange (i.e. transpiration and net assimilation) with a WALZ GFS-3000 photosynthesis system (Walz, Effeltrich, Germany), equipped with a lamp, set to a light intensity of 1000 µmol m$^{-1}$ s$^{-1}$ at ambient air temperature and humidity. $CO_2$ was set to 400 ppm, close to the ambient $CO_2$ level. Gas exchange rates were further calibrated to the actual leaf area by scanning measured leaves to determine projected leaf area. Gas exchange measurements were accompanied by leaf water potential ($\Psi_l$) on one to three leaves from each sapling with the Scholander pressure chamber technique (*Scholander et al., 1965*). Leafy branchlets of 5–6 cm length were cut from all the trees in the experiments and put in a pressure chamber (PMS Instrument, Albany, OR) fed by a nitrogen gas cylinder. Gas pressure within the chamber was gradually increased (~1 MPa min$^{-1}$) until water emerged from the protruding cut branch surface, and the negative value of the pressure was recorded as water potential ($\Psi_l$) in MPa.

## Sampling root exudates

Root exudates were collected from intact fine roots using a non-soil syringe system modified from *Phillips et al., 2008*. Root tips were sampled from the side windows of the Rhizobox, and were remained attached to the trees during the entire procedure until harvest. We gently washed the intact fine roots using a spray bottle, with autoclaved carbon-free nutrient solution (0.5 mM $NH_4NO_3$, 0.1 mM $KH_2PO_4$, 0.2 mM $K_2SO_4$, 0.2 mM $MgSO_4$, 0.3 mM $CaCl_2$) and fine forceps to remove soil particles and other possible contaminants. The fine roots were placed into a 20 mL sterile plastic syringe and filled with 0.5–1.3 mm acid-washed glass beads and 10 mL autoclaved carbon-free nutrient solution. Then, the syringes were covered with aluminum foil and covered with soil to block sunlight and heat. After 48 hr, the nutrient solution was collected from each syringe system. An additional 10 mL of the double distilled water was flushed through the syringe system to obtain a representative carbon recovery. Two samples and one control (solution subjected to the same process without a root; metabolites that were found in control samples were not determined as exudate metabolites) per Rhizobox were included. Small carbon amounts that were found in the control (root-free) tubes were regarded as contamination and were subtracted from the carbon amounts in the samples. At the end of incubation, the two samples were pooled together and then separated into two tubes, one for exudation rate and the other for metabolomics analysis (below). All solutions were filtered immediately through a 0.22 µm sterile syringe filter (Millex PVDF, Millipore Co., Billerica, MA) and stored in the lab at –80°C until analysis. The solutions were analyzed for dissolved organic carbon on a total organic carbon analyzer (Aurora 1030 W TOC Analyzer coupled with a 1088 rotatory TOC auto sampler; OI Analytical, TX). Root exudation rates were calculated as the total amount of carbon flushed from the pooled root system over the incubation period divided by root dry weight of the investigated root strand, and hereafter referred to as specific exudation rate (µg C mg root$^{-1}$ day$^{-1}$). After root exudate collection, roots were cut off the tree and dried by oven (48 hr, 60°C) and then weighed.

## Sample preparation for metabolite profiling and metabolite extraction

Aliquots of root exudates were prepared based on the (lower) amount of organic carbon (aforementioned TOC measurement) in the control samples, in order to normalize for variations in quantity among samples. Aliquots were freeze-dried by a lyophilizer. Extraction and analysis were performed as previously described (*Dong et al., 2016*), with some modifications: lyophilized exudates and controls were extracted with 1 mL of a pre-cooled (–20°C) homogenous methanol:methyl-tertbutyl-ether (TMBE) 1:3 (v/v) mixture containing the following internal standards: 0.1 µg mL$^{-1}$ of phosphatidylcholine 34:0 (17:0/17:0) and 0.15 nmol mL$^{-1}$ of LM6002 (Avanti) standard mix. The tubes were vortexed and then sonicated for 30 min in ice-cold sonication bath (taken for a brief vortex every 10 min). Then, UPLC grade water:methanol (3:1, v/v) solution (0.5 mL) containing internal standards ($^{13}$C and $^{15}$N labeled amino acids standard mix; Sigma) was added to the tubes followed by centrifugation. The upper organic phase was transferred into a 2 mL Eppendorf tube. The polar phase was re-extracted as described above, with 0.5 mL of TMBE. Both organic phases were combined and dried in speedvac and then stored at –80°C until analysis. The lower, polar phase used for polar metabolite analysis was stored at –80°C until analysis.

## LC-MS polar metabolites analysis and identification

Metabolic profiling of polar phase was done as previously described (*Zhang et al., 2016*) with minor modifications described below. Briefly, analysis was performed using Acquity I class UPLC System combined with mass spectrometer (Thermo Exactive Plus Orbitrap; Waltham, MA). The mass spectrometer was operated under the following parameters: full MS/dd-MS2 mode (1 μscans) at 35,000 resolution from 75 to 1050 m/z, with 3.25 kV spray voltage, 40 sheath gas, 10 auxiliary gas and negative ionization mode. The LC separation was done using the SeQuant Zic-pHilic (150 mm × 2.1 mm) with the SeQuant guard column (20 mm × 2.1 mm) (Merck; Kenilworth, NJ). The mobile phase A was 20 mM ammonium carbonate plus 0.1% ammonia hydroxide in water and the mobile phase B was acetonitrile. The flow rate was kept at 200 μL min$^{-1}$ and the gradient was as follows: 0–2 min 75% of B, 14 min 25% of B, 18 min 25% of B, 19 min 75% of B, for 4 min. The data processing was done using TraceFinder (Thermo Fisher software), where detected compounds were identified by retention time and fragments were verified using in-house mass spectra library. The results were normalized using the internal standards peak area.

## LC-MS semi-polar metabolites analysis and identification

Metabolic profiling of semi-polar phase was performed using Waters ACQUITY UPLC system coupled to a Vion IMS qToF mass spectrometer (Waters Corp., Milford, MA). The LC separation was performed as previously described (*Itkin et al., 2011*) with the minor modifications described below. Briefly, the chromatographic separation was performed on an ACQUITY UPLC BEH C18 column (2.1×100 mm, i.d., 1.7 μm) (Waters Corp., Milford, MA). The mobile phase A consisted of 95% water (UPLC grade) and 5% acetonitrile, with 0.1% formic acid; mobile phase B consisted of 100% acetonitrile with 0.1% formic acid. The column was maintained at 35°C and flow rate of the mobile phase was 0.3 mL min$^{-1}$. Mobile phase A was initially run at 100%, and it was gradually reduced to 72% at 22 min, following a decrease to 0% at 36 min. Then, mobile phase B was run at 100% until 38 min; next, mobile phase A was set to 100% at 38.5 min. Finally, the column was equilibrated at 100% mobile phase A until 40 min. MS parameters were as follows: the source and de-solvation temperatures were maintained at 120°C and 350°C, respectively. The capillary voltage was set to 2 and 1 kV at negative and positive ionization mode, respectively; cone voltage was set at 40 V. Nitrogen was used as de-solvation gas and cone gas at the flow rate of 700 and 50 L h$^{-1}$, respectively. The mass spectrometer was operated in full scan HDMS$^E$ negative or positive resolution mode over a mass range of 50–2000 Da. For the high-energy scan function, a collision energy ramp of 20–80 eV was applied, and for the low-energy scan function 4 eV was applied. Leucine-enkephalin was used as lock-mass reference standard. LC-MS data were analyzed and processed with UNIFI (Version 1.9.4, Waters Corp., Milford, MA). The putative identification of the different semi-polar species was performed by comparison accurate mass, fragmentation pattern, and ion mobility (CCS) values to an in-house semi-polar database, where several compounds were identified vs. standards, when available. Several compounds were identified by their theoretical fragmentation, accurate mass, and CCS.

## Leaf elements quantification

*C. sempervirens* leaves from each tree were collected at the time points indicated above. They were set to dry in an oven at 60°C for 48 hr and ground to obtain a fine powder. Next, 0.1 g leaf powder was weighed and burned to ash at 550°C for 6 hr. Next, 0.5 mL of fresh nitric acid (HNO$_3$ 69%; Merck) was added and incubated for 5 days (*Laursen et al., 2009*). Subsequently, 19.5 mL of deionized water (Millipore, Milli-Q Biocel Water Purification System, Germany) was added and filtered. The elements quantification in the leaves was performed using inductively coupled plasma mass spectrometry (ICP-MS; Agilent 7500ce; Agilent Technologies, Wokingham, UK) tuned in standard mode. The plasma power was operated at 1450±50 W and the argon carrier and make-up gases were set at 0.83 and 0.17 L min$^{-1}$, respectively. Sample uptake was maintained at ca. 0.1 mL min$^{-1}$ by a self-aspirating perfluoroalkoxy micro-flow nebulizer (Agilent Technologies). Elimination of spectral interferences was obtained by the use of an octopole ion guide with the cell gases helium or hydrogen (*Laursen et al., 2009*). For the series of 235 *C. sempervirens* samples, three replicates of certified reference material NIST 1575a (Pine Needles) and NIST 1547 (Peach leaves) were included. Only data deviating <±10% from the certified reference values were retained.

## Leaf and root element mapping

Elemental mapping of leaf and root samples was performed using an X-ray fluorescence microscope (AttoMap; Sigray, San Francisco, CA) at the company's labs. The system uses an ultra-high brightness X-ray source with multiple X-ray targets to provide optimal detection sensitivity; a novel X-ray optics providing large focused X-ray flux, small focus (<10 μm) and large working distance (>25 mm); and a high speed detector. Elemental distribution images (heat maps) were created at 20 μm resolution and scan area of 500 μm with a copper target to detect Fe, Mn, Ca, K, and Cl, and molybdenum or gold target to detect Cu and Zn.

## Statistical analysis

Due to the multidisciplinary nature of the study, statistical analysis was designed separately for the plant physiology and microbial inoculation sections, and for the metabolomics and bacterial growth sections. The two major factors tested in our experiment were drought (treated vs. irrigated saplings) and bacterial inoculation (with and without). All measurements were compared to the baseline samples in the beginning of the experiment, except for the CFU. Effects on root exudation rates were assessed by means of two-way ANOVA. Effects on physiological parameters, tree biomass, soil and leaf elements, and CFU values were assessed by means of a two-way repeated measures ANOVA using the general linear model procedure in using Origin 7 (Origin Lab Corporation, Northampton, MA). ANOVA was followed by Bonferroni tests, and Tukey's HSD test to determine where the significant difference lay within the dataset. p-Value < 0.05 was taken to be significant. Data presented are the means of at least six replicates (from independent saplings) unless otherwise stated. Thereafter, technical replicates were done while including these controls in the experiment, to ensure reproducibility.

Metabolome values were assessed by means of two-way ANOVA, followed by post hoc Tukey's test with FDR correction and p-value adjustment for multiple comparisons. Statistically different masses, comparing those from irrigated and drought-exposed saplings, with and without bacterial inoculations, were assigned to metabolites (*Figure 3—source data 1 and 2*). Then, the chemical annotation of significantly different metabolites was inspected manually and classified according to four levels of confidence of metabolite identification as previously described (*Dunn et al., 2013*; *Sumner et al., 2007*; *Schymanski et al., 2014*). PCoA was performed using MetaboAnalyst V4.0, and the metabolites analysis was performed using R and the interface RStudio. Bacterial growth curves were constructed in Matlab (MathWorks, Natick, MA), and presented as means of four biological replications with standard deviations.

## Acknowledgements

We thank The Weizmann Tree Lab members for support, advice, and helpful discussions throughout. We also thank Guy Shmuel for providing us with a protocol of ICP-MS; Nir Galili for help with earlier experiments; Roee Ben Nissan for assistance with programing of bacteria growth curve; Ron Rotkopf for guidance with statistical analysis; and Sergey Kapishnikov for coordinating the elemental mapping experiments (Weizmann Institute of Science). We thank SH Lau and the engineers of Sigray (San Francisco, CA) for performing the elemental mapping on our samples; Robert Fluhr (Weizman Institute of Science), Daniel Dar (Caltech, CA), and Sophie Obersteiner (Ben Gurion University, Israel), for providing helpful comments on earlier versions of this paper. Funding: The project was funded by The Edith and Nathan Goldenberg Career Development Chair; Mary and Tom Beck-Canadian Center for Alternative Energy Research; Larson Charitable Foundation New Scientist Fund; Angel Faivovich Foundation for Ecological Research; Yotam Project; Dana and Yossie Hollander; Estate of Emile Mimran; and Estate of Helen Nichunsky.

# Additional information

## Funding

| Funder | Grant reference number | Author |
|---|---|---|
| Edith and Natan Goldenberg Career Development Chair | | Tamir Klein |
| Mary and Tom Beck | Canadian Center for Alternative Energy Research | Tamir Klein |
| Larson Charitable Foundation New Scientist Fund | | Tamir Klein |
| Angel mFaivovich Foundation for Ecological Research | | Tamir Klein |
| Yotam project | | Tamir Klein |
| Dana and Yossie Hollander | | Tamir Klein |
| Estate of Emile Mimran | | Tamir Klein |
| Estate of Helen Nichunsky | | Tamir Klein |

The funders had no role in study design, data collection and interpretation, or the decision to submit the work for publication.

## Author contributions

Yaara Oppenheimer-Shaanan, Formal analysis, Methodology, Writing - original draft; Gilad Jakoby, Maxim Itkin, Sergey Malitsky, Formal analysis, Methodology; Maya L Starr, Romiel Karliner, Gal Eilon, Methodology; Tamir Klein, Conceptualization, Supervision, Funding acquisition, Writing - original draft, Writing - review and editing

## Author ORCIDs

Yaara Oppenheimer-Shaanan (iD) http://orcid.org/0000-0002-7005-3074
Maxim Itkin (iD) http://orcid.org/0000-0003-1348-2814
Tamir Klein (iD) http://orcid.org/0000-0002-3882-8845

## Decision letter and Author response

Decision letter https://doi.org/10.7554/eLife.79679.sa1
Author response https://doi.org/10.7554/eLife.79679.sa2

# Additional files

## Supplementary files
• MDAR checklist

## Data availability

All data related to the study are reported in the manuscript and Supplementary Information. Source Data files 1-6 are included in the submission.

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
