## [Editor Report]

This paper will be of interest to those interested in plant-microbe interactions under drought. Trees can exchange root exudates for minerals with soil bacteria. In a pot experiment under realistic conditions, the authors indicate that this exchange persists, and may be protective, when plants experience drought stress. The combination of methods used is an important strength: visualizing bacterial colonization of roots, measuring root exudation and mineral uptake, in conjunction with separate assays of bacterial growth responses to chemicals released by trees.

---

## [Decision Letter]

**Decision letter after peer review:**

[Editors’ note: the authors submitted for reconsideration following the decision after peer review. What follows is the decision letter after the first round of review.]

Thank you for submitting the paper "A dynamic rhizosphere interplay between tree roots and soil bacteria under drought" for consideration by *eLife*. Your article has been reviewed by 3 peer reviewers, and the evaluation has been overseen by a Reviewing Editor and a Senior Editor. The following individuals involved in review of your submission have agreed to reveal their identity: Alexander Weinhold (Reviewer #1); Richard Phillips (Reviewer #3).

Comments to the Authors:

We are sorry to say that, after consultation with the reviewers, we have decided that this work will not be considered further for publication by *eLife* in its current form. If the concerns raised by the reviewers can be addressed, the study could be considered for publication in *eLife*, but the current requests go beyond what we would consider under revisions.

Specifically, important strengths of this study are its attempt to investigate root exudate composition while manipulating native root-microbe interactions and drought stress, in order to draw conclusions about ecological roles of root exudates in relevant scenarios, while furthermore using methods originating in different disciplines for the appropriate manipulation and assessment of experimental factors. The topic is important and many aspects of the experimental design well-chosen, especially considering ecological relevance of the system and the potential power of factorial design, and such a study could be of interest for *eLife*. Measuring metabolites released from roots and connecting them to their putative functions is extremely difficult, even more so when done with trees and actual soil. Yet all reviewers identified substantial flaws in the analysis and presentation of the results, and independently of these concerns, it is difficult to connect the various results to the authors' stated research questions. Thus, the analysis is inconclusive, and it is generally not clear what we learn out of this study regarding trees under drought stress engaged in interactions with soil bacteria. This is reflected in the divergent focus of the individual reviews, each of which should provide very helpful suggestions for addressing these concerns. In addition to the points raised by the reviewers (below), I would note the following: (1) It is not clear how the bacteria were chosen, except that these are two native soil strains, which is commendable – but are they among the most common, most frequently found associated with the study plant, … ? (2) Metabolite naming is inconsistent, e.g. Figure 7 refers to 3,4-DHB which I think is in fact 3,4-DHM, a supplementary figure refers to nicotine which I think is nicotinate. (3) Heat map colors are confusing, indicating up-regulation (red) and down-regulation (blue) when in fact everything is expressed as a positive fold change, which would be best shown in darker (highest) to lighter (lowest) colors. (4) The first results show that, while roots are recruiting, the experiment is carried out with an unstable microbial community, which makes it more challenging to interpret the results (stability in the experimental microcosms is achieved only for the re-inoculated irrigation treatment). (5) It is misleading to refer e.g. to "drought-specific" root exudate metabolites, as Figure 5 shows that all exudate metabolites are detected across all treatments even if they may have different patterns in each treatment (hard to tell from the color scheme in Figure 5 – difference between slightly darker and slightly lighter yellow hard to assess).

*Reviewer #1 (Recommendations for the authors):*

Oppenheimer-Shaanan et al. investigated the how inoculation with root bacteria can changes the ability of the Mediterranean cypress to cope with drought stress. Therefore, they compared root exudation (as TOC) of tree saplings under drought stress and under constant irrigation. They show that root exudation increases when trees are inoculated with root bacteria. Inoculated trees also showed higher capabilities to recover from drought stress.

The combination of the different methodologies provides an interesting view, or glimpse as the authors wrote, on the interaction between tree roots and associated bacteria under drought stress. The use of the rhizoboxes, the collection of root exudates and the analysis of the bacteria is very nice experimental set up. It allows to see the different effect of abiotic stress on the different players in the interaction (soil, roots, bacteria, leaves) and to make connections on how they are interacting.

However, I see here some shortcomings in the analysis of the data and the presentation. Especially, the metabolomics data are not easy to understand and could be much better presented. Furthermore, I think that the emphasis that is put on the fold changes of certain metabolites is too high. From this are conclusions drawn that are not well supported by the data and makes it hard to follow later the discussion.

Condensation and maybe a different statistical analysis in some figures in the metabolomics analysis would help to better visualize the changes and to see clearer the effects of bacteria and drought.

I don't feel qualified to review the bacterial stain isolation, characterization and construction.

I am not used so much to the use of those log2 fold changes. That is why I need to do some calculations. If I did those right most of the log2FC in Table 1 are very low there and all smaller than 0.5/-0.5 that means a fold change of less than 1.5 I am wondering if those will have a physiological effect. You than showed the effect of some of the metabolites like in FIGURE 6, where you are testing p- coumaric acid for bacterial growth. In the source data the fold change for this metabolite is either 1.09 or 1.15 only (if I calculate back from your log2 value). I think that is very low and I have my doubts that you can draw some conclusion from that. I acknowledge the effects that you see in bacteria assays, and I don't doubt that. But I just think that the difference in concentrations would not matter. How would this look in a comparison where you for instance test different concentrations of e.g. p-coumaric acid and see if the drought or irrigated concentrations have different effects.

I have also some difficulties with the interpretation and analysis of the metabolomics data. The PCA is a good way of visualizing the differences between treatment groups, but I think a PCoA would be better here. It is getting more and more used in metabolomics research and has to advantage to deal better with non-normal distributed data, which you will most likely have. In combination with the PCoA a PERMANOVA might help you to show effects of the treatment instead of the visual comparison. That would also make either the heatmaps in Figure 5 more understandable. For me, the fold change and their origins are not clear not clear. In source data 4 there is no metabolite 25fold more abundant. So, I am not sure where this data is coming from.

One suggestion would also be to combine the Figure 5 and 6 in one figure, maybe similar to Figure 4. That would make it much easier for the reader to compare the drought and re-irrigation effects.

I like the Figure 1. I am just wondering if this would be better at the end of paper as summary. Also, I can't really see in the figure the differences between the different treatments etc. Also, the (5) is missing in the legend. Wouldn't it be good to show the changes, for instance when you say something is inhibited that the arrow is brocken etc? (Just some artistic suggestion).

In general, I also feel that the source data need to be better referenced in the text (e.g. source data 4 sheet E) Also it needs to be better explained. Source data 3, for instance, needs more explanation on what the two different fold changes columns are. Because if this is used in Figure 4, then I don't know where the data in the boxplot are coming from. When I am looking Figure 4 and see the change of re irrigated TOC. I don't see how this is significant, because the boxplots are within then quantile of the other one. It would help to see the source data.

Line 100ff this sentence comes very unexpected and may need some further explanation.

Line 111: That is introduced a bit to far away. Can you test the microbial priming in your experiment?

Line 124-129 I like the idea of having this results summary at the beginning of the result section.

Line 141: I would put this biomass data in a table or supplemental data. Later in the discussion it was very difficult to find again the data in the text.

Line 152: How was the relative abundance calculated?

Line 125/ Figure 2: The 10fold change would be easier to see with log scale, in Figure 3 you use them.

Line160 – 162: Why is that? How can they be helping to overcome the drought when they are so decreasing in their abundance.

Figure 4 legend refers to abundance in Figure 1 (that is the overview should be Figure 2,3).

Line 202: "…Their exudation was enriched again" I can’t see this in Figure 4 B, this figure is about TOC not single metabolites.

Line 205 How do you know that exudates are mainly secondary metabolites? Where is this shown? How do you classify them? The same in Line 208f. Where does this statement come from? How do you know the percentage and what do you mean of total metabolites (detected, identified, extracted)?

Line 212 What is a feature? That was not explained before. I know it, but do other readers know too?

Line 215ff Where is the data for the chemical richness? It is not in Figure s4B also chemical richness is never explained, it is also not in Source Data 5.

Line 221f there is no figure S7C.

Line 325f those are the wrong figures cited. Again, how do you know that mostly secondary metabolites were exuded. I don't see the data to support this.

Line 336 No, always the inoculated have higher weights. Line 141ff.

Line 354 Debatable with the data and scale shown, if both Figures would be in log scale. Also, the raw data would be nice.

Line 401 where did you identify a salicylic acid

Line 669 the MS parameters for the polar analysis are missing

*Reviewer #2 (Recommendations for the authors):*

Trees can exchange root exudates for minerals with soil bacteria. In a pot experiment the authors show that this exchange was enhanced when water was withheld, suggesting that trees can recruit beneficial bacteria under drought conditions. The experiment factorially combined the irrigation vs. drought treatment with no inoculation vs. inoculation of pots with two bacterial species. There were strong effects of the bacterial inoculation, but it was not directly tested if they were stronger under drought or irrigation (this could be tested by the interaction term in 2-way anovas). A strength of this study is that the authors measured root exudates, bacterial abundance and concentrations of soil P and minerals during and after the drought. To which extent the results of this short-term experiment with saplings of one particular tree species and two particular bacterial species in reasonably large pots may be extrapolated would have to be tested with further experiments or comparative field studies.

As indicated above, there is one major issue that makes it difficult to judge this manuscript. This is that the main hypotheses concern a potential interaction between the two treatment factors drought and inoculation, yet the authors do not test this interaction statistically. It is not possible to draw conclusions from separate tests of inoculation under irrigation and drought because one cannot draw conclusions about the significance of an interaction from separate tests. Thus, if one test is just significant and the other just not significant, the interaction may be very far from significant.

It should be very easy for the authors to use full general linear models for ALL their measurements (no need to use different analyses for different measurements except for the PCA type of analyses). These can go beyond the mentioned factorial 2-way anova with interaction and also include e.g. TOC as covariate for the analysis of Figure 4, fitting the covariate both as main term and as interaction with the treatment terms and their interaction. One could even add some path-analytic model to test the causal hypotheses implicit in the interpretation of the results.

On issue that did not become clear to me was if the 6 saplings per treatment combination were individually planted in 6 separate boxes (pots) or if perhaps two or three were in one box. My concern stems from looking at Table 2, where pots with four tree species are mentioned, so I wondered why it was not mentioned for the main experiment if each sapling was planted separately into a box. If there were fewer boxes than saplings in the main experiment, then box would have to be added as a random term in the general linear models.

Besides this main concern I find the description of results and the discussion complicated and on the long side. I was also confused that often only three treatments seemed to be compared: irrigated, drought without inoculation and drought with inoculation. For example, in Figures5 and 6 one can see that results should have been available for irrigated without inoculation and irrigated with inoculation, but then these are not always shown (e.g. Figure 5A). When tests are mentioned of irrigation vs. drought it is not even clear if means across the inoculation treatments were used or not.

There are some language issues that the authors should resolve, as well as some minor glitches such as "Schleppi et al. 2020 missing in the reference list.

*Reviewer #3 (Recommendations for the authors):*

In "A dynamic rhizosphere interplay between tree roots and soil bacteria under drought", the authors present compelling evidence that trees exposed to drought can alter their rhizosphere environment by (1) altering their exudation profiles (amount and composition) and (2) promoting bacteria that can increase the availability of soil nutrients such as phosphorus. The authors used a nice combination of experiments – collecting and analyzing the root exudates from potted saplings of Mediterranean cypress exposed to drought, adding root-associated bacteria with fluorescent markers, and adding exudate compounds to bacterial cultures, and determining the plant physiological responses to bacterial additions and drought. The authors found that while root exudation was decreased by drought, select rhizosphere microbes that were likely promoted by the tree's root exudation profile, buffering the trees from the some of the nutritional consequences of the drought by allowing them to sustain levels of nutrient uptake.

Overall, the study was carried out carefully, and there are several novel aspects. First, while there have been some studies that have looked at how root exudation changes in response to environmental stress or the presence of select microbes, few have combined the two to look at the interactive effects of microbes and stress. Second, few studies have characterized the metabolites released by roots and even fewer have conducted assays to examine how exuded metabolites impact bacterial growth and the consequences of this for plant nutrition. The exudation part of the story alone would have been an excellent contribution to the literature and by making the connections between the root exudates and the bacterial activity, the authors have provided a nice blueprint for how future studies might explore the costs/benefits of plant-microbe interactions.

Nevertheless, there are several issues that if addressed, would strengthen the manuscript. These include:

Improved clarity in writing. There are several places where the writing is unclear and editing would be helpful. In the line-by-line comments, I have included many suggested edits, especially for the Introduction. I didn't provide these throughout the manuscript (in order to focus on other aspects of the manuscript) but I suggest the authors pay careful attention to this issue in all sections of the manuscript.

Revising tables and figures. Overall, several display items could be improved. In my view, Figure 1 does not offer much and fails to capture the factorial nature of the experiment. What about replacing this with a box and arrow diagram that shows the how the presence of microbes likely alters tree drought tolerance (sort of like a path analysis but with words)? For Figure 2, rather than showing the relative abundances (panel B), it would be helpful to see the absolute changes in the two bacterial taxa in C (as opposed to the combined "with bacteria" treatment). If you do decide to include this, you could still keep the figures simple by removing the time series from the figure are interesting, it's not a major part of the story so perhaps that could be moved to SI? For Figure 3, the same comment applies. For Figure 4, it's unclear how a correlation coefficient can be calculated from the data collected. If the exudation rates are generated from root systems that contain both bacterial taxa, how can you have different relationships? For Figure 8b, the treatments were a bit confusing since the color scheme is the same as Figure 4a but the treatments are different.

In Table 2, I assume K in the last column refers to potassium (it's not in the legend). Also, please use consistent units for the soil variables, as some are in mE per L whereas others are mg per kg soil. In the SI, Figure S3 would benefit from having the inoculation and watering treatments added as vertical lines or you could add the sharing and arrows used in S3 to S2 so that the two figures are consistent. Finally, readers are likely to find it easier to make comparisons across figures if the time series data are presented as days of the month (May, 21, June 16, etc.) be as opposed to DD/MM.

There are few points not covered in the manuscripts (or barely covered) that could strengthen the manuscript. First, there's little discussion of the differences between active and passive exudation. Some roots exude compounds merely because a concentration gradient exists between the root apoplast and the soil solution (passive) whereas some exudates are actively exuded. In the case of a drought-stressed root, how much of the exudation is merely of the roots physical integrity? A desiccated rot likely lacks physical integrity to keep exudates from leaking out and it seems like an investigation of how the roots themselves changed with the treatments (especially the drought) is warranted.

Additionally, the manuscript would benefit from more synthesis about what is known about root exudation under drought. There have been many studies of tree root exudation (either solution culture or pulse labeling) that have shown how and why exudation rates change under drought. A brief synthesis of these papers would be very helpful to the readers. Also, I encourage the authors to calculate the percentage of photo assimilate that was exuded (in all four treatments) so that comparisons can be made with other exudation studies.

Finally, while the readers will appreciate the known caveat that no mycorrhizal fungi were detected in the pots, some discussion of how mycorrhiza fungi could affect exudation and bacterial growth would be extremely useful. In forests, arbuscular mycorrhizal fungi (which associate with cypress) play a central role in P nutrition, tree drought tolerance and root-microbial interactions. So, it's worth speculating how some of the patterns and processes described in the study might be altered (even if all you can do is speculate) if the fungi had been present.

Line by line comments:

L1. You might consider using a title that highlights one of the studies main findings: "Altered root exudation during drought stimulates rhizosphere bacteria, with consequences for tree drought tolerance".

L19. "minerals" should be "mineral nutrients" or just "nutrients".

L21. I would be more careful here; exudates can affect nutrient availability directly (e.g.. solubilizing mineral-P) independent of their effects on bacteria.

L22. "Priming" typically refers to the accelerated decomposition of organic matter owing to C (or in some cases N) inputs. In the case of P, phosphates are esther-bonded and so the release of P from soil organic matter is not considered priming. See Figure 2 in Dijkstra et al. 2013 "Rhizosphere priming: a nutrient perspective" for a discussion of this.

L25. Readers may be confused by calling this a factorial design without more details. The two bacterial taxa were not included as individual treatments since they were combined into a "with bacteria" treatment.

L28. State percentage increase of exudation?

L30. Would be clearer to readers if you started a new sentence here. "In a second experiment, we added to bacterial cultures metabolites detected as root exudates, and found that xx percent stimulated bacterial growth".

L34. Could add (recruitment of beneficial bacteria) "especially under water stress".

L39. Maybe just start with "Climate change".

L41. Change to "…ecosystems worldwide, with negative impacts for forest health".

L43-44. Please list references for each impact (recruitment, mortality, etc.)

L47. A closing sentence is needed here.

L49. Replace with "including escape, avoidance and tolerance strategies".

L52. Replace with "Root systems mediate water and nutrient uptake…".

L55. Not sure about the relevance of mentioning fine root turnover here since it's not the focus of the paper.

L49-84. There are lots of redundancies in paragraphs two and three of the Introduction. The two might be better as a single paragraph about what is known about plant exudation responses to drought, and the impacts of these exudate effects on microbes.

L86-89. Is one of these sentences the topic sentence of the paragraph? Please try to make this clear so that the readers know what the paragraph is about from the outset.

L101. As noted above, priming refers to something beyond just root exudates stimulation in that it refers to the accelerated decay of soil organic matter. Some people refer to what you're describing as "microbial activation" which is a hypothesis of how the first step of priming occurs.

L106. "This increase in exudation occurred despite increases in photosynthesis throughout the dry season."

L111 and L118. Replace "priming" with another term please. "exudate-induced microbial growth".

L124-129. This "results abstract" does not see necessary unless the journal specifically requires it.

L140. In the droughted trees in Figure S2, the bacteria treatment seems to be reducing assimilation rates relative to the no bacteria treatment. What might be causing this? This finding seems to be at odds with the other data you present that shows that the presence of bacteria in the drought treatment help trees maintain their nutritional status. Is there some tradeoff that night be occurring which would cause assimilation to go down?

L141. Replace "sapling" with "saplings".

L142-144. Please present a p value for this contrast of a statistical test was conducted.

L146-162. Were statistical tests conducted? If yes, please present the results. If not, please explain why?

L171. It would be helpful if you could use this format – presenting the % difference in exudation – for the other treatments.

L174. How is a correlation coefficient calculated when you have one root system (which are used to generate exudation data) but both types of bacteria? Please explain what variables are being used.

L186. Figure 5B?

L202. Is this meant to be Figure 5B (and not 4B)?

L290. Replace "C source" with "C uptake" or "C assimilation"?

L298. But lower assimilation in the bacteria-drought treatment (as least based on Figure S2).

L311. Replace "sustain" with "remain"?

L365. The was these references are used in this sentence makes it seem like the studies were of tree roots (but they were not).

L486-489. Was any data collected on the changes in PAR over the course of the experiment? Were trees in full sun?

L633. Do you have data on what C contamination existed in the non-root controls? Was this subtracted from the values of the ones with roots (i.e. were the fluxes corrected for background interferences)? Some info in the challenges of getting a C-free root chamber would be helpful to readers who want to repeat these methods.

[Editors’ note: further revisions were suggested prior to acceptance, as described below.]

Thank you for resubmitting your work entitled "A dynamic rhizosphere interplay between tree roots and soil bacteria under drought stress" for further consideration by *eLife*. Your revised article has been evaluated by Meredith Schuman (Senior Editor) and a Reviewing Editor.

The manuscript has been improved but there are some remaining issues that need to be addressed, as outlined below:

Essential revisions:

(1) *eLife* has a consultative review procedure. Subsequent to the submission of both reviews, we discussed Figure 2 due to the Editor's concerns. This figure presents data which are very important for interpretation of the authors' proposed mechanism. In the current form of the manuscript the interpretation of these data is misleading.

(1a) These data show that drought-stressed trees release less TOC (exudate) from roots than irrigated trees, and that both drought-stressed and irrigated trees release more TOC when inoculated. In contrast, trees re-irrigated after drought produce similar TOC from roots whether they are inoculated or not (although the figure indicates there is also a significant difference there, which is confusing and perhaps incorrectly indicated). Thus, the inference is rather that inoculated plants produce more TOC from roots, except if they have been subjected to drought and re-irrigation. What the authors imply is that drought-stressed plants produce more exudates when inoculated, which is correct but misleading since irrigated plants do as well. It is rather interesting that re-irrigated plants do not do this. Reviewer 1 commented additionally: "This contrasts with the answer of the authors to my main question: 'Thus, only under drought conditions were trees responsive to inoculation, showing a ~50% increase in root exudation (Figure 2; P = 0.039 for the interaction irrigation:bacteria).' I really cannot see this interaction in Figure 2, and if anything, it seems to go into the wrong direction (i.e. difference with-without bacterial is larger under irrigation)."

(1b) The exudate production has no predictable correlation with abundance of either of the two bacterial strains monitored by the authors. The reason for choosing these strains is now clear in the text, but there is no indication that either responds to differences in root exudates. Thus, although the growth of both strains is affected by (some of) the exudate metabolites which differ (mildly – less than 2x) in abundance between treatments, there is no evidence that these two strains are directly associated with exudate differences. The inoculation affects total TOC from roots, but the effect could be indirect in the ecologically relevant, but complex soil environment of the experiment.

(1c) There are no data to indicate that these bacterial strains are responsible for the nutrient effects observed for inoculated plants undergoing drought, especially as the soil for the experiment was not sterilized. The unsterilized soil is in fact a nice aspect of the design from an ecological perspective, but also makes it more complicated to determine the mechanism by which additional inoculation has its effects.

2) Reviewer 1 emphasized that the writing still needs substantial correction and streamlining.

The full reviews are provided below. Please note that the topics under (1) arose after the reviews were submitted and during the consultation.

*Reviewer #1 (Recommendations for the authors):*

This manuscript has been improved during revision, including improved statistical analysis. In my view the Discussion would still benefit from a better focus on the main results.

I mentioned in the previous review that parts of the paper were quite long and complicated to read and still find this to be true (see e.g. Discussion lines 413-443; overall, the Discussion has become even longer than it was before). I also mentioned that there were many small language errors and again can still find them in this "revised" version. It is really the authors responsibility to correct the language because they are the "publisher".

---

## [Author Response]

[Editors’ note: the authors resubmitted a revised version of the paper for consideration. What follows is the authors’ response to the first round of review.]

Comments to the Authors:Specifically, important strengths of this study are its attempt to investigate root exudate composition while manipulating native root-microbe interactions and drought stress, in order to draw conclusions about ecological roles of root exudates in relevant scenarios, while furthermore using methods originating in different disciplines for the appropriate manipulation and assessment of experimental factors. The topic is important and many aspects of the experimental design well-chosen, especially considering ecological relevance of the system and the potential power of factorial design, and such a study could be of interest for eLife. Measuring metabolites released from roots and connecting them to their putative functions is extremely difficult, even more so when done with trees and actual soil. Yet all reviewers identified substantial flaws in the analysis and presentation of the results, and independently of these concerns, it is difficult to connect the various results to the authors' stated research questions. Thus, the analysis is inconclusive, and it is generally not clear what we learn out of this study regarding trees under drought stress engaged in interactions with soil bacteria. This is reflected in the divergent focus of the individual reviews, each of which should provide very helpful suggestions for addressing these concerns. In addition to the points raised by the reviewers (below), I would note the following:1) It is not clear how the bacteria were chosen, except that these are two native soil strains, which is commendable – but are they among the most common, most frequently found associated with the study plant, …?

We thank you for this insightful comment. We now add information in the

Methods to clarify our choice: “In multiple screens of rhizosphere bacteria in our forest site, a total of 54 bacterial strains were identified on plates. Of which, *B. subtilis* and *P. stutzeri* were the most consistent, evidencing their important role in our system”.

In addition, we modified the text with the first mention of the bacteria in the Results: “Fluorescently tagged *B. subtilis* and *P. stutzeri*, that were modified from native strains isolated from the forest soil, showed attachment and dispersed colonization along *Cupressus* fine roots, regardless of irrigation, on days 1 and 3 following inoculation (Figure 2A, Source Data 2)”.

2) Metabolite naming is inconsistent, e.g. Figure 7 refers to 3,4-DHB which I think is in fact 3,4-DHM, a supplementary figure refers to nicotine which I think is nicotinate.

Thank you so much for catching this error with 3,4-DHM, which we have now corrected. Regarding nicotinate, we added a clarification in the Methods: “Metabolites were identical with those identified in root exudates (Table 1), except for nicotinate, which was tested in the form of nicotine”.

3) Heat map colors are confusing, indicating up-regulation (red) and down-regulation (blue) when in fact everything is expressed as a positive fold change, which would be best shown in darker (highest) to lighter (lowest) colors.

We understand the Editor’s point and hence tried a single-tone heat map, which, to us, was disappointing, since differences among metabolites and among treatments were hardly visible. Therefore, we would like to keep our heat map colors. It is very common to use contrasting colors for different levels of metabolites; the heat map shows the changes of the metabolite levels relative to the beginning of the experiment.

4) The first results show that, while roots are recruiting, the experiment is carried out with an unstable microbial community, which makes it more challenging to interpret the results (stability in the experimental microcosms is achieved only for the re-inoculated irrigation treatment).

We appreciate the Editor’s comment. We are aware that the natural populations of bacteria vary according to the different treatments, but we wanted to focus on the bacteria we added to the soil. Our conclusions from the experiment relate to *Bacillus subtilis* and *Pseudomonas stutzeri* addition to the soil, so we also tested the specific metabolites on the growth of these bacteria under laboratory conditions. The study aimed to show the dynamic changes between these specific bacteria and the cypress roots under dry conditions. The dynamics in the rhizosphere and soil during the experiment were affected by both the native microbial community and bacteria inoculation. We did not follow the general microbial community in the rhizosphere and soil in this experiment. Instead, we monitored only the inoculation bacteria by isolating them from the soil by a selective medium and counting them. We point to the dynamics of these two bacteria and not to the whole microbial community. To address this comment, we added text in the “study limitations” section of the Discussion: “Second, manipulating the rhizosphere with inoculations can also be criticized as creating disturbance. Moreover, except for the re-inoculated irrigated trees, bacterial communities were not stable. Still, the two bacteria species were released into the forest soil to compete as any other microbial species in the soil system”. In addition, a sentence was added in the Results to clarify our approach: “The inoculations can be regarded as pulses, allowing us to test the rhizosphere interactions, while declining with time”.

(5) It is misleading to refer e.g. to "drought-specific" root exudate metabolites, as Figure 5 shows that all exudate metabolites are detected across all treatments even if they may have different patterns in each treatment (hard to tell from the color scheme in Figure 5 – difference between slightly darker and slightly lighter yellow hard to assess).

Following the Editor’s comment, we changed the term "drought-specific" to “drought increased” in.

Reviewer #1 (Recommendations for the authors):Oppenheimer-Shaanan et al. investigated the how inoculation with root bacteria can changes the ability of the Mediterranean cypress to cope with drought stress. Therefore, they compared root exudation (as TOC) of tree saplings under drought stress and under constant irrigation. They show that root exudation increases when trees are inoculated with root bacteria. Inoculated trees also showed higher capabilities to recover from drought stress.The combination of the different methodologies provides an interesting view, or glimpse as the authors wrote, on the interaction between tree roots and associated bacteria under drought stress. The use of the rhizoboxes , the collection of root exudates and the analysis of the bacteria is very nice experimental set up. It allows to see the different effect of abiotic stress on the different players in the interaction (soil, roots, bacteria, leaves) and to make connections on how they are interacting.However, I see here some shortcomings in the analysis of the data and the presentation. Especially, the metabolomics data are not easy to understand and could be much better presented. Furthermore, I think that the emphasis that is put on the fold changes of certain metabolites is too high. From this are conclusions drawn that are not well supported by the data and makes it hard to follow later the discussion.Condensation and maybe a different statistical analysis in some figures in the metabolomics analysis would help to better visualize the changes and to see clearer the effects of bacteria and drought.

We thank reviewer #1for taking the time to thoroughly review our work. We have rigorously revised the manuscript following the valuable suggestions of the reviewer. This includes; Figures have been merged and moved to streamline the narrative; The text was shortened in the Introduction and Results; Instability of the microbial communities was discussed under the study limitations; Information was added concerning the saplings’ biomass; PCA of the polar metabolite profiles of exudates from drought-exposed saplings with and without bacterial inoculations was replaced with PCoA. Please see below detailed responses.

I don't feel qualified to review the bacterial stain isolation, characterization and construction.I am not used so much to the use of those log2 fold changes. That is why I need to do some calculations. If I did those right most of the log2FC in Table 1 are very low there and all smaller than 0.5/-0.5 that means a fold change of less than 1.5 I am wondering if those will have a physiological effect. You than showed the effect of some of the metabolites like in FIGURE 6, where you are testing p- coumaric acid for bacterial growth. In the source data the fold change for this metabolite is either 1.09 or 1.15 only (if I calculate back from your log2 value). I think that is very low and I have my doubts that you can draw some conclusion from that. I acknowledge the effects that you see in bacteria assays, and I don't doubt that. But I just think that the difference in concentrations would not matter. How would this look in a comparison where you for instance test different concentrations of e.g. p-coumaric acid and see if the drought or irrigated concentrations have different effects.

We thank reviewer #1 for taking the time to thoroughly review our work. First, we note that these are the log 2 fold changes from the pairwise comparison between two treatments, not the changes from the control baseline, which were much higher. A reference to our approach can be found in Warren (2016). Second, fold changes between the treatments might be relatively small, yet significant between treatments (Table 1). Third, small molecules tend to be found at low concentrations in root exudates and may be rapidly consumed by rhizosphere microbes, which affect their recovery and analysis. Here we showed growth curves of the bacteria where very low concentrations of these molecules had physiological effects. Indeed, a small difference in concentration can have a large effect, as observed e.g. in Williams et al. (2011).

Warren, C. R. (2016). Simultaneous efflux and uptake of metabolites by roots of wheat. *Plant and Soil*, *406*(1), 359-374.

Williams, A., Langridge, H., Straathof, A. L., Fox, G., Muhammadali, H., Hollywood, K. A., … and de Vries, F. T. (2021). Comparing root exudate collection techniques: An improved hybrid method. *Soil Biology and Biochemistry*, *161*, 108391.

I have also some difficulties with the interpretation and analysis of the metabolomics data. The PCA is a good way of visualizing the differences between treatment groups, but I think a PCoA would be better here. It is getting more and more used in metabolomics research and has to advantage to deal better with non-normal distributed data, which you will most likely have. In combination with the PCoA a PERMANOVA might help you to show effects of the treatment instead of the visual comparison. That would also make either the heatmaps in Figure 5 more understandable. For me, the fold change and their origins are not clear not clear. In source data 4 there is no metabolite 25fold more abundant. So, I am not sure where this data is coming from.

We thank the reviewer for this suggestion. To address this reviewer’s point, we performed PCoA and PERMANOVA and found similar trends to those observed by the PCA. These changes are reflected in the new Figure 3A and in the Results text: “Principal coordinate analysis (PCoA) showed that root exudates blends (samples) from drought trees that were inoculated and those from drought trees that were not inoculated partitioned into distinct clusters, with partial overlap (Figure 3A). When samples from irrigated trees were also considered in the PCoA, they behaved similarly, however there was no single sample common to all three groups (Figure 3A)”.

One suggestion would also be to combine the Figure 5 and 6 in one figure, maybe similar to Figure 4. That would make it much easier for the reader to compare the drought and re-irrigation effects.

We thank the reviewer for this suggestion, which we adopted.

I like the Figure 1. I am just wondering if this would be better at the end of paper as summary. Also, I can't really see in the figure the differences between the different treatments etc. Also, the (5) is missing in the legend. Wouldn't it be good to show the changes, for instance when you say something is inhibited that the arrow is brocken etc? (Just some artistic suggestion).

Thank you for this suggestion. We have restructured the figure and moved it to the end of the manuscript. We also added the reference to step (5).

In general, I also feel that the source data need to be better referenced in the text (e.g. source data 4 sheet E) Also it needs to be better explained. Source data 3, for instance, needs more explanation on what the two different fold changes columns are. Because if this is used in Figure 4, then I don't know where the data in the boxplot are coming from. When I am looking Figure 4 and see the change of re irrigated TOC. I don't see how this is significant, because the boxplots are within then quantile of the other one. It would help to see the source data.

We have addressed this point by providing further data of TOC in Source Data 3 and by providing a new supplementary file with raw data of TOC and assimilation rates. The analyses of the total amount of carbon were done compared to the zero time of the experiment. At that time, samples were also taken, and all the experimental groups were under similar conditions. There are slight differences between the individual trees in the experiment from the beginning. Therefore, the measurements describe changes between the start point and a subsequent measurements (fold changes) for each individual tree. Only then were statistical analyses done.

The significant differences are compared between two groups and plotted on the graph were examined in statistical tests that are in the Source Data 3. We claim for significant differences between the groups that do not overlap for example between irrigated trees with and without bacteria.

Line 100ff this sentence comes very unexpected and may need some further explanation.

We add explanation: “In the forest soil, root exudation is suspected to enhance rhizobacteria, in turn leading to increased decomposition of soil organic matter, i.e., increased C mineralization, a process termed microbial priming (Schleppi et al., 2019). Here, we use this term on a wide perspective, even that P release from soil phosphates, for example, is not strictly considered as priming (Dijkstra et al. 2013)”.

Line 111: That is introduced a bit too far away. Can you test the microbial priming in your experiment?

We clarified this sentence and toned it down: “Our overall objective was to test whether microbial recruitment by tree root exudates (a form of microbial priming) is beneficial to trees under drought, an abiotic stress that alters tree carbon and nutrient allocation”.

Line 141: I would put this biomass data in a table or supplemental data. Later in the discussion it was very difficult to find again the data in the text.

Biomass data were added in Source Data 1, and the text was modified: “During the experiment, sapling biomass increment was 188.6±15.6 g and 149.1±14.4 g for irrigated saplings with and without bacterial inoculations, respectively (difference not significant; *P* = 0.753; Source Data 1). The effect of drought was highly significant (*P* < 0.001), and biomass increment in drought-exposed saplings with and without bacterial inoculations was 22.4±5.7 g and 4.5±4.6 g, respectively (*P* = 0.09)”.

Line 152: How was the relative abundance calculated?

This information belongs to the Methods: “The relative abundance was estimated by counting each strain of bacteria separately and diving by the total amount of bacteria.”

Line 125/ Figure 2: the 10fold change would be easier to see with log scale, in Figure 3 you use them.

We modified Figure 2.

Line160 – 162: Why is that? How can they be helping to overcome the drought when they are so decreasing in their abundance.

Bacteria in the rhizosphere are even more critical during the drought, even at small amounts. Our study shows this, in agreement with earlier works (Shakya et al., 2013; Taketani et al., 2016, Yang et al., 2021). We show that bacteria adapted to dry conditions (*B. subtilis*) can improve tree nutrition to higher extent. This claim was also discussed in the Discussion. To address the comment about the unstable bacterial communities, we added text in the “study limitations” section of the Discussion: “Second, manipulating the rhizosphere with inoculations can also be criticized as creating disturbance. Moreover, except for the reinoculated irrigated trees, bacterial communities were not stable. Still, the two bacteria species were released into the forest soil to compete as any other microbial species in the soil system”. In addition, a sentence was added in the Results to clarify our approach: “The inoculations can be regarded as pulses, allowing us to test the rhizosphere interactions, while declining with time”.

Yang, N., Nesme, J., Røder, H.L. et al. Emergent bacterial community properties induce enhanced drought tolerance in Arabidopsis. npj Biofilms Microbiomes 7, 82 (2021).

Figure 4 legend refers to abundance in Figure 1 (that is the overview should be Figure 2,3).

Thank you so much for catching these confusing errors, which we have now corrected.

Line 202: "…Their exudation was enriched again" I can’t see this in Figure 4 B, this figure is about TOC not single metabolites.

Thank you for catching these errors, which we have now corrected. The reference was changed to Figure 3B (the old Figure 6).

Line 205 How do you know that exudates are mainly secondary metabolites? Where is this shown? How do you classify them? The same in Line 208f. Where does this statement come from? How do you know the percentage and what do you mean of total metabolites (detected, identified, extracted)?

Point well taken. We did chemical classification analysis which can found in Source Data 4E, which was now referenced in this section. This is similar with earlier studies like e.g. Gargallo-Garriga et al. (2018). The percentages relate to the identified metabolites with LC-MS that generated a diverse profile of secondary metabolites.

Gargallo-Garriga, A., Preece, C., Sardans, J., Oravec, M., Urban, O., and Peñuelas, J. (2018). Root exudate metabolomes change under drought and show limited capacity for recovery. *Scientific reports*, *8*(1), 1-15.

Line 212 What is a feature? That was not explained before. I know it, but do other readers know too?

Following the reviewer’s comment we changed the text to: “clustered mass signals” to clarify the sentence.

Line 215ff Where is the data for the chemical richness? It is not in Figure s4B also chemical richness is never explained, it is also not in Source Data 5.

The sentence was removed.

Line 221f there is no figure S7C.Line 325f those are the wrong figures cited. Again, how do you know that mostly secondary metabolites were exuded. I don't see the data to support this.

Thank you so much for catching these errors, which we have now corrected.

Line 336 No, always the inoculated have higher weights. Line 141ff.

Thank you, this has now been corrected: “Finally, desiccated trees supplemented with soil bacteria showed better nutrition in P and Fe (Figure 5), and higher biomass (although not significantly); but, unexpectedly, did not recover faster than without bacteria. Thus, photosynthesis recovered faster in drought trees that were not inoculated than those that were (Figure S3)”.

Line 354 Debatable with the data and scale shown, if both Figures would be in log scale. Also, the raw data would be nice.

Following the reviewer’s comment, we are adding another supplementary file with statistical analysis of root exudation rates and leaf assimilation rates across all trees along the experiment. This analysis is now cited at the respective sections in the Discussion: “root exudation increased following inoculation, and was higher during drought than following it (Source Data 3, 7)”.

Line 401 where did you identify a salicylic acid

We identified 4-AMINOSALICYLIC ACID (See Source Data 4A) as one of the metabolites but not significant to bacteria inoculation.

Line 669 the MS parameters for the polar analysis are missing

Thank you so much for catching these missing. We added the missing details: “Briefly, analysis was performed using Acquity I class UPLC System combined with mass spectrometer (Thermo Exactive Plus Orbitrap; Waltham, MA, USA). The mass spectrometer was operated under the following parameters: Full MS/ dd-MS2 mode (1 μscans) at 35,000 resolution from 75-1050 m/z, with 3.25 kV spray voltage, 40 sheath gas, 10 auxiliary gas and negative ionization mode”.

Reviewer #2 (Recommendations for the authors):Trees can exchange root exudates for minerals with soil bacteria. In a pot experiment the authors show that this exchange was enhanced when water was withheld, suggesting that trees can recruit beneficial bacteria under drought conditions. The experiment factorially combined the irrigation vs. drought treatment with no inoculation vs. inoculation of pots with two bacterial species. There were strong effects of the bacterial inoculation, but it was not directly tested if they were stronger under drought or irrigation (this could be tested by the interaction term in 2-way anovas). A strength of this study is that the authors measured root exudates, bacterial abundance and concentrations of soil P and minerals during and after the drought. To which extent the results of this short-term experiment with saplings of one particular tree species and two particular bacterial species in reasonably large pots may be extrapolated would have to be tested with further experiments or comparative field studies.As indicated above, there is one major issue that makes it difficult to judge this manuscript. This is that the main hypotheses concern a potential interaction between the two treatment factors drought and inoculation, yet the authors do not test this interaction statistically. It is not possible to draw conclusions from separate tests of inoculation under irrigation and drought because one cannot draw conclusions about the significance of an interaction from separate tests. Thus, if one test is just significant and the other just not significant, the interaction may be very far from significant.It should be very easy for the authors to use full general linear models for ALL their measurements (no need to use different analyses for different measurements except for the PCA type of analyses). These can go beyond the mentioned factorial 2-way anova with interaction and also include e.g. TOC as covariate for the analysis of Figure 4, fitting the covariate both as main term and as interaction with the treatment terms and their interaction. One could even add some path-analytic model to test the causal hypotheses implicit in the interpretation of the results.

We thank the reviewer for the enthusiastic feedback. Following up on the reviewer’s suggestion, two-way ANOVA was analyzed to check the different interactions

between the treatments (indicated in the table), which is indicated in the text. We also changed the one-way ANOVA of metabolomes values to two-way ANOVA. We re-analyzed the treatments’ effects on the physiological measurements, bacterial counts, TOC, metabolites (polar and semi polar) and the amounts of leaf elements. Please see all the results in source data files. This issue is now highlighted throughout the Results chapters: “Bacterial inoculations had no effect on leaf gas exchange, and hence the interaction irrigation:bacteria was not significant either”. “Thus, only under drought conditions were trees responsive to inoculation, showing a ~50% increase in root exudation (Figure 2; *P* = 0.039 for the interaction irrigation:bacteria)”. “In two metabolites, the interaction irrigation:bacteria was significant under drought, but not under re-irrigation (Source Data 4)”. The Methods section was updated too. In addition, following the reviewer’s proposal, a general linear model was tested concerning one of the physiological parameters, namely leaf assimilation, where there was a sharp difference between drought treatment and irrigation, and the exudation rate (total organic carbon). Please see Source Data 7. The differences were similar to those from the analysis done without the linear model. The linear model supported the correlation and indicated the strength of a causal hypothesis, but did not prove the direction of causation. We were unable to use TOC as an independent (predictor) in a linear model. The TOC was dependent (criterion) on drought/irrigation and bacteria inoculation treatments, and these interactions were examined. Finally, following a suggestion by another reviewer, we changed the PCA graphs with PCoA, which resulted with similar results.

On issue that did not become clear to me was if the 6 saplings per treatment combination were individually planted in 6 separate boxes (pots) or if perhaps two or three were in one box. My concern stems from looking at Table 2, where pots with four tree species are mentioned, so I wondered why it was not mentioned for the main experiment if each sapling was planted separately into a box. If there were fewer boxes than saplings in the main experiment, then box would have to be added as a random term in the general linear models.

Thank you, we found your comment helpful and have revised accordingly. First, the erroneous title of Table 2 was corrected: “Soil properties from pots where saplings were grown under treatments of drought or irrigated with or without bacteria”. Second, our experiment was conducted with each sapling being planted in an individual rhizobox. So, we added this information in the Methods text: “The saplings were kept at the net-house, one sapling per rhizobox, under optimal irrigation of 1.2 L day^-1^ for a seven months period, to allow root growth into mixed the forest soil”.

Besides this main concern I find the description of results and the discussion complicated and on the long side. I was also confused that often only three treatments seemed to be compared: irrigated, drought without inoculation and drought with inoculation. For example, in Figures5 and 6 one can see that results should have been available for irrigated without inoculation and irrigated with inoculation, but then these are not always shown (e.g. Figure 5A). When tests are mentioned of irrigation vs. drought it is not even clear if means across the inoculation treatments were used or not.

We apologize for the length and complexity of our presentation and have tried to improve in multiple places along the text and figures. The opening paragraph of the Results was completely removed. We also condensed and merged paragraphs 2 and 3 of the Introduction into a single paragraph. We were able to better streamline the Results and Discussion by moving Figure 1 to the end; merging Figures 2 and 3; and merging Figures 5 and 6. In the new Figure 3, we compare only the treatments showing significant differences in the total organic carbon.

We add more explanation in to clarify the treatments which were tested. For example, Figure 3: (A) and (B) Metabolic profiles at drought period, (C) at re-Irrigation period. In addition, clarifications were added in text, e.g. in Results: “The amount of organic carbon exuded from roots of drought-exposed trees was significantly, ~50% lower than from roots of irrigated trees, and moreover, ~70% lower, in the re-irrigation period (across inoculation treatments; *P* < 0.01; Figure 2, Source Data 3, Supplementary Information)”.

There are some language issues that the authors should resolve, as well as some minor glitches such as "Schleppi et al. 2020 missing in the reference list.

English errors were corrected throughout the text. The missing reference was added.

Reviewer #3 (Recommendations for the authors):In "A dynamic rhizosphere interplay between tree roots and soil bacteria under drought", the authors present compelling evidence that trees exposed to drought can alter their rhizosphere environment by (1) altering their exudation profiles (amount and composition) and (2) promoting bacteria that can increase the availability of soil nutrients such as phosphorus. The authors used a nice combination of experiments – collecting and analyzing the root exudates from potted saplings of Mediterranean cypress exposed to drought, adding root-associated bacteria with fluorescent markers, and adding exudate compounds to bacterial cultures, and determining the plant physiological responses to bacterial additions and drought. The authors found that while root exudation was decreased by drought, select rhizosphere microbes that were likely promoted by the tree's root exudation profile, buffering the trees from the some of the nutritional consequences of the drought by allowing them to sustain levels of nutrient uptake.Overall, the study was carried out carefully, and there are several novel aspects. First, while there have been some studies that have looked at how root exudation changes in response to environmental stress or the presence of select microbes, few have combined the two to look at the interactive effects of microbes and stress. Second, few studies have characterized the metabolites released by roots and even fewer have conducted assays to examine how exuded metabolites impact bacterial growth and the consequences of this for plant nutrition. The exudation part of the story alone would have been an excellent contribution to the literature and by making the connections between the root exudates and the bacterial activity, the authors have provided a nice blueprint for how future studies might explore the costs/benefits of plant-microbe interactions.Nevertheless, there are several issues that if addressed, would strengthen the manuscript. These include:Improved clarity in writing. There are several places where the writing is unclear, and editing would be helpful. In the line-by-line comments, I have included many suggested edits, especially for the Introduction. I didn't provide these throughout the manuscript (in order to focus on other aspects of the manuscript) but I suggest the authors pay careful attention to this issue in all sections of the manuscript.Revising tables and figures. Overall, several display items could be improved. In my view, Figure 1 does not offer much and fails to capture the factorial nature of the experiment. What about replacing this with a box and arrow diagram that shows the how the presence of microbes likely alters tree drought tolerance (sort of like a path analysis but with words)? For Figure 2, rather than showing the relative abundances (panel B), it would be helpful to see the absolute changes in the two bacterial taxa in C (as opposed to the combined "with bacteria" treatment). If you do decide to include this, you could still keep the figures simple by removing the time series from the figure are interesting, it's not a major part of the story so perhaps that could be moved to SI? For Figure 3, the same comment applies. For Figure 4, it's unclear how a correlation coefficient can be calculated from the data collected. If the exudation rates are generated from root systems that contain both bacterial taxa, how can you have different relationships?

We thank reviewer #3 for his thorough review. We are glad that he found our results interesting and novel, and we thank him for the many insightful comments that contribute to improving our manuscript. Following up on the above suggestions, Figure 1 was moved to the end of the figures (now Figure 7), where it better fits the overview of our experiment. We have also modified it slightly to improve clarity, yet we are hesitant to present a box and arrow diagram, which might look too deterministic, considering that our results are coming from a pot experiment on saplings. Figures 2 and 3 (now Figure 1) were modified as suggested, with absolute changes in the two bacterial taxa. Regarding Figure 4 (now Figure 2), indeed we had a single exudate mean per treatment and time, and two bacterial abundance values, one for each of the two species. While interactions between the populations of the two strains may exist, it is still possible to correlate between each population and the exudate rates. In addition, Figures 5 and 6 were merged into the new Figure 3.

For Figure 8b, the treatments were a bit confusing since the color scheme is the same as Figure 4a but the treatments are different.

We now add more details in the caption for clarity: “Leaf elements (A) and soil phosphorous (B) of irrigated (blue shades) and drought-exposed and re-irrigated (brown shades) *Cupressus sempervirens* saplings, with and without bacterial inoculations”.

In Table 2, I assume K in the last column refers to potassium (it's not in the legend). Also, please use consistent units for the soil variables, as some are in mE per L whereas others are mg per kg soil.

Thank you for this comment. We modified the Table caption and harmonized the units as suggested, to clarify this point.

In the SI, Figure S3 would benefit from having the inoculation and watering treatments added as vertical lines or you could add the sharing and arrows used in S3 to S2 so that the two figures are consistent.

We added to Figure S3 the same lines as in Figure S2. The caption was updated accordingly.

Finally, readers are likely to find it easier to make comparisons across figures if the time series data are presented as days of the month (May, 21, June 16, etc.) be as opposed to DD/MM.There are few points not covered in the manuscripts (or barely covered) that could strengthen the manuscript. First, there's little discussion of the differences between active and passive exudation. Some roots exude compounds merely because a concentration gradient exists between the root apoplast and the soil solution (passive) whereas some exudates are actively exuded. In the case of a drought-stressed root, how much of the exudation is merely of the roots physical integrity? A desiccated rot likely lacks physical integrity to keep exudates from leaking out and it seems like an investigation of how the roots themselves changed with the treatments (especially the drought) is warranted.

This is an interesting point that deserves more discussion. Here, we were able to include it in new text that was introduced to the section on study limitations: “passive exudation and leakage from drought-stressed roots cannot be ruled out in our experiment.

However, the observed shift in metabolites and the sensitivity to inoculations support rather an active exudation”.

Additionally, the manuscript would benefit from more synthesis about what is known about root exudation under drought. There have been many studies of tree root exudation (either solution culture or pulse labeling) that have shown how and why exudation rates change under drought. A brief synthesis of these papers would be very helpful to the readers.

Point well taken. Per the reviewer’s comment we improved the focus of our relevant Introduction section, and added a key reference on tree root exudation changes under drought: “The majority of root exudates typically consists of primary metabolites (20%; sugars, amino acids, and organic acids) and 15% of nitrogen as well as secondary metabolites, complex polymers, such as flavonoids, glucosinolates, auxins, etc. (Vives-Peris et al., 2020). Those plant-derived metabolites were shown to shape microbial communities by allowing bacteria to metabolize them and then establish themselves in the rhizosphere (Venturi, 2016; Sasse et al., 2017). Although root exudation is ubiquitous among tree species, the amount and composition of root exudates vary. So far, little information is available on how drought influences tree root exudates, their chemical composition, and how root metabolism is connected with shifts in root-associated microbiome composition (Zhang et al., 2007; Tückmant el al., 2017; Naylor and Coleman-Derr 2018). Recently, it was shown that oak trees (*Quercus ilex*) under drought shift their exudates from primary to secondary metabolites (Gargallo-Garriga et al. 2018)”.

We also used this information in the Discussion: “Root exudates included carbohydrates and organic acids that fed bacterial communities (Figures 3, 4). Yet, unexpectedly, most root exudates under drought were secondary, rather than primary metabolites. However, this is in agreement with a similar shift observed in the exudate metabolomes of drought-exposed oak trees (Gargallo-Garriga et al. 2018). Indeed, phenolic acid compounds and amino acid derivatives proved to be superior carbon and nitrogen sources than a sugar and amino acid”.

Gargallo-Garriga, A., Preece, C., Sardans, J., Oravec, M., Urban, O., and Peñuelas, J. (2018). Root exudate metabolomes change under drought and show limited capacity for recovery. *Scientific reports*, *8*(1), 1-15.

Also, I encourage the authors to calculate the percentage of photo assimilate that was exuded (in all four treatments) so that comparisons can be made with other exudation studies.

Good point. While true percentages would require a full carbon balance at the whole sapling scale (which was not done here), ratios could be readily calculated. Text was added in the Results section on root exudates: “An additional measure of tree carbon allocation into root exudation is the ratio between exudation rate (µg C mg root^-1^ day^-1^) and net assimilation rate (µmol C m^-2^ leaf s^-1^). This ratio was 0.40-0.45 under drought, decreasing to 0.27-0.35 under re-irrigation (across inoculation treatments). In irrigated saplings, ratios increased from 0.12-0.17 without bacteria, to 0.34 and 0.58 in inoculated saplings (in later and earlier phases of the experiment, respectively)”.

Finally, while the readers will appreciate the known caveat that no mycorrhizal fungi were detected in the pots, some discussion of how mycorrhiza fungi could affect exudation and bacterial growth would be extremely useful. In forests, arbuscular mycorrhizal fungi (which associate with cypress) play a central role in P nutrition, tree drought tolerance and root-microbial interactions. So it's worth speculating how some of the patterns and processes described in the study might be altered (even if all you can do is speculate) if the fungi had been present.

We thank for the reviewer for this important point. To account for this point, text was added under study limitations: “In the forest, mycorrhizal fungi, which were missing here, can have large effects too (Meier et al. 2013). Specifically, *Cupressus sempervirens* hosts a diversity of active arbuscular mycorrhizal species (Avital et al. 2022). While the latter generally help in tree P nutrition, their activity is mostly reduced under drought. Future studies should test the functions of the three-kingdom interactions among trees, fungi and bacteria”.

Meier, I. C., Avis, P. G., and Phillips, R. P. (2013). Fungal communities influence root exudation rates in pine seedlings. *FEMS microbiology ecology*, *83*(3), 585-595. Avital S, Rog I, Livne-Luzon S, Cahanovitc R, Klein T (2022) Asymmetric belowground carbon transfer in a diverse tree community. Molecular Ecology.

Rog, I., Tague, C., Jakoby, G., Megidish, S., Yaakobi, A., Wagner, Y., and Klein, T. (2021).

Interspecific soil water partitioning as a driver of increased productivity in a diverse mixed Mediterranean forest. *Journal of Geophysical Research: Biogeosciences*, *126*(9), e2021JG006382.

Line by line comments:L1. You might consider using a title that highlights one of the studies main findings: "Altered root exudation during drought stimulates rhizosphere bacteria, with consequences for tree drought tolerance".

We thank reviewer for his suggestion. While such a title captures one of the main findings, it misses others, and thus we prefer to retain out title.

L19. "minerals" should be "mineral nutrients" or just "nutrients".

Done.

L21. I would be more careful here; exudates can affect nutrient availability directly (e.g.. solubilizing mineral-P) independent of their effects on bacteria.

Thank you, we modified the sentence “However, root exudates typically decrease in situations such as drought, calling into question the efficacy of solvation and bacteria dependent mineral uptake in such stress”.

L22. "Priming" typically refers to the accelerated decomposition of organic matter owing to C (or in some cases N) inputs. In the case of P, phosphates are esther-bonded and so the release of P from soil organic matter is not considered priming. See Figure 2 in Dijkstra et al. 2013 "Rhizosphere priming: a nutrient perspective" for a discussion of this.

Thank you for your insightful comment. Although the abstract is too condensed to include such details, we do find this point highly relevant, and hence corrected for it in the Introduction: “In the forest soil, root exudation is suspected to enhance rhizobacteria, in turn leading to increased decomposition of soil organic matter, i.e., increased C mineralization, a process termed microbial priming (Schleppi et al., 2019). Here, we use this term on a wide perspective, even that P release from soil phosphates, for example, is not strictly considered as priming (Dijkstra et al. 2013)”. We then come back to this point in the Discussion: “The carbon cost of root exudation has already been accounted in relation to P foraging (Dijkstra et al. 2013, Wang and Lambers 2020), where rhizodeposition may be used for P scavenging rather than for decomposition of oil organic matter. On the other hand, slower recovery might indicate a more controlled acclimation response (Bastida et al. 2019, Gessler et al. 2020)”

Bastida, F., García, C., Fierer, N. et al. Global ecological predictors of the soil priming effect. Nat Commun 10, 3481 (2019).

Dijkstra Feike, Carrillo Yolima, Pendall Elise, Morgan Jack. (2013) Rhizosphere priming: a nutrient perspective. Frontiers in Microbiology 4.

L25. Readers may be confused by calling this a factorial design without more details. The two bacterial taxa were not included as individual treatments since they were combined into a "with bacteria" treatment.

Thank you for the comment, we change the text in line 25 “A 1-month imposed drought and concomitant inoculations with a mix of *Bacillus subtilis* and *Pseudomonas stutzeri*, bacteria species isolated from the forest soil, were applied using factorial design”.

L28. State percentage increase of exudation?

Yes, this was now added: “Interestingly, root exudation rates increased 2.3-fold with bacteria under drought”.

L30. Would be clearer to readers if you started a new sentence here. "In a second experiment, we added to bacterial cultures metabolites detected as root exudates, and found that xx percent stimulated bacterial growth".

Thank you, the text was modified as suggested: “Forty four metabolites in exudates were significantly different in concentration between irrigated and drought trees, including phenolic acid compounds and quinate. When adding these metabolites as carbon and nitrogen sources to bacterial cultures of both bacterial species, 8 of 9 metabolites stimulated bacterial growth”.

L34. Could add (recruitment of beneficial bacteria) "especially under water stress".L39. Maybe just start with "Climate change".L41. Change to "…ecosystems worldwide, with negative impacts for forest health".

Corrected as suggested.

L43-44. Please list references for each impact (recruitment, mortality, etc.)

References were added: “Drought adversely affects forest health in many aspects, including seedling recruitment (Pozner et al. 2022), tree productivity (Klein *et al.*, 2014), and mortality of trees (Klein *et al.*, 2019), with increased susceptibility to pathogen or insect attack (Reichstein *et al.*, 2013; McDowell et al. 2022)”.

Pozner, E., Bar-On, P., Livne-Luzon, S., Moran, U., Tsamir-Rimon, M., Dener, E., … and Klein, T. (2022). A hidden mechanism of forest loss under climate change: The role of drought in eliminating forest regeneration at the edge of its distribution. *Forest Ecology and Management*, *506*, 119966.

McDowell, N. G., Sapes, G., Pivovaroff, A., Adams, H. D., Allen, C. D., Anderegg, W. R., … and Xu, C. (2022). Mechanisms of woody-plant mortality under rising drought, CO2 and vapour pressure deficit. *Nature Reviews Earth and Environment*, 1-15.

L47. A closing sentence is needed here.

We added: “The capacity of trees to survive future droughts is virtually unknown (McDowell et al. 2022).”

L49. Replace with "including escape, avoidance and tolerance strategies".L52. Replace with "Root systems mediate water and nutrient uptake…".

Done.

L55. Not sure about the relevance of mentioning fine root turnover here since it's not the focus of the paper.

Yes, but we wish to provide the full context of our study.

L49-84. There are lots of redundancies in paragraphs two and three of the Introduction. The two might be better as a single paragraph about what is known about plant exudation responses to drought, and the impacts of these exudate effects on microbes.

We have condensed the text and merged these paragraphs into a single one: “Trees have evolved mechanisms to cope with drought, including escape, avoidance and tolerance strategies. Drought has been shown to induce an alteration of carbon allocation from aboveground to below ground organs and increase in the amounts of soluble sugars in the roots (Huang et al., 2000; Hasibeder et al., 2015). Root systems mediate water and nutrient uptake, provide physical stabilization, store nutrients and carbohydrates, and provide carbon and nutrients to the soil through the process of fine-root turnover (Brunner and Godbold, 2007; Haichar et al. 2008; Ryan, 2011; Harfouche 2014; Jarzyniak and Jasiński, 2014; Klein 2016). In addition to these roles, trees invest a substantial part of their photosynthesized carbon into root exudates that entice and presumably feed plant-beneficial and root associated microbiota (Bais et al., 2006; Badri and Vivanco 2009; Karst et al., 2017; Jakoby et al., 2020). In parallel, the rhizosphere microbes can promote plant growth through various mechanisms such as increasing the availability of nutrients, secreting phytohormones, suppressing pathogens, or having positive effects on the plant metabolism (Perez-Montano et al., 2014; Zhou et al., 2015). Microbial utilization and metabolism play a central role in modulating concentration gradients of a variety of compounds right outside root tips, thereby constituting a soil sink (Dakora and Phillips, 2002; Mommer et al., 2016; Martin et al., 2017; Tsonuda and van Dam 2017). The majority of root exudates typically consists of primary metabolites (20%; sugars, amino acids, and organic acids) and 15% of nitrogen as well as secondary metabolites, complex polymers, such as flavonoids, glucosinolates, auxins, etc. (Vives-Peris et al., 2020). Those plant-derived metabolites were shown to shape microbial communities by allowing bacteria to metabolize them and then establish themselves in the rhizosphere (Venturi, 2016; Sasse et al., 2017). Although root exudation is ubiquitous among tree species, the amount and composition of root exudates vary. So far, little information is available on how drought influences tree root exudates, their chemical composition, and how root metabolism is connected with shifts in root-associated microbiome composition (Zhang et al., 2007; Tückmant el al., 2017; Naylor and Coleman-Derr 2018). Recently, it was shown that oak trees (*Quercus ilex*) under drought shift their exudates from primary to secondary metabolites (Gargallo-Garriga et al. 2018)”.

L86-89. Is one of these sentences the topic sentence of the paragraph? Please try to make this clear so that the readers know what the paragraph is about from the outset.

We admit that the structure was confusing here, and hence reordered these sentences: “The chemical composition of root exudates have a direct effect on the rhizosphere communities. These include plant growth-promoting rhizobacteria (PGPR) genera such as *Bacillus, Pseudomonas, Enterobacter, Acinetobacter, Burkholderia, Arthrobacter,* and *Paenibacillus* (Sasse et al., 2017; Zhang et al., 2017). For example, the banana root exudate fumaric acid attracts the Gram-positive *Bacillus subtilis* N11 and stimulates biofilm formation (Zhang *et al.*, 2014)”.

L101. As noted above, priming refers to something beyond just root exudates stimulation in that it refers to the accelerated decay of soil organic matter. Some people refer to what you're describing as "microbial activation" which is a hypothesis of how the first step of priming occurs.

The text has been modified to reflect these points: “In the forest soil, root exudation is suspected to enhance rhizobacteria, in turn leading to increased decomposition of soil organic matter, i.e., increased C mineralization, a process termed microbial priming (Schleppi et al., 2019). Here, we use this term on a wide perspective, even that P release from soil phosphates, for example, is not strictly considered as priming (Dijkstra et al. 2013)”.

L106 "This increase in exudation occurred despite increases in photosynthesis throughout the dry season."

Following the reviewer’s comment the text was changed: “This increase in exudation occurred in spite of the sharp decrease in photosynthesis throughout the dry season, and more so in the coniferous *Cupressus sempervirens*”.

L111 and L118. Replace "priming" with another term please. "exudate-induced microbial growth".

Following the reviewer’s comment the text was clarified in both locations: “Our overall objective was to test whether microbial recruitment by tree root exudates (a form of microbial priming) is beneficial to trees under drought, an abiotic stress that alters tree carbon and nutrient allocation” and “Our major hypothesis regarded the existence of an exudate induced microbial-tree interaction cycle, starting with tree stress and subsequent exudation, on to enhancement of soil bacteria and their activity, and back to improved tree nutrition”.

L124-129. This "results abstract" does not see necessary unless the journal specifically requires it.

Per the reviewer’s comment, the section was removed.

L140. In the droughted trees in Figure S2, the bacteria treatment seems to be reducing assimilation rates relative to the no bacteria treatment. What might be causing this? This finding seems to be at odds with the other data you present that shows that the presence of bacteria in the drought treatment help trees maintain their nutritional status. Is there some tradeoff that night be occurring which would cause assimilation to go down?

True. This observation is discussed in the third paragraph in the Discussion, which has now been expanded: “Finally, desiccated trees supplemented with soil bacteria showed better nutrition in P and Fe (Figure 5), and higher biomass (although not significantly); but, unexpectedly, did not recover faster than without bacteria. Thus, photosynthesis recovered faster in drought trees that were not inoculated than those that were (Figure S3). Considering that inoculated trees invested significantly more carbon into the rhizosphere than uninoculated drought trees, it is possible that this carbon cost came at the expense of internal tree reserves, later creating a ‘recovery penalty’ for the trees (Gessler et al. 2020). The carbon cost of root exudation has already been accounted in relation to P foraging (Dijkstra et al. 2013, Wang and Lambers 2020), where rhizodeposition may be used for P scavenging rather than for decomposition of oil organic matter”.

L141. Replace "sapling" with "saplings".

Done.

L142-144. Please present a p value for this contrast of a statistical test was conducted.

Biomass data were added in Source Data 1, and the text was modified: “During the experiment, sapling biomass increment was 188.6±15.6 g and 149.1±14.4 g for irrigated saplings with and without bacterial inoculations, respectively (difference not significant; *P* = 0.753; Source Data 1). The effect of drought was highly significant (*P* < 0.001), and biomass increment in drought-exposed saplings with and without bacterial inoculations was 22.4±5.7 g and 4.5±4.6 g, respectively (*P* = 0.09)”.

L146-162. Were statistical tests conducted? If yes, please present the results. If not, please explain why?

Thank you for the comment. Statistical results were added to the text: “Differences between days in relative abundance were significant (*P* < 0.001; Source Data 2)”. In addition, in: “Bacterial abundance was also lower in drought trees than around irrigated (Figure 1B) or reirrigated trees (Figure 1C) (*P* < 0.001; Source Data 2)”.

L171. It would be helpful if you could use this format – presenting the % difference in exudation – for the other treatments.

The text was modified as suggested: “The amount of organic carbon exuded from roots of drought-exposed trees was significantly, ~50% lower than from roots of irrigated trees, and moreover, ~70% lower, in the re-irrigation period (across inoculation treatments; *P* < 0.01; Figure 2, Source Data 3, Supplementary Information). Under constant irrigation, bacterial inoculation significantly increased root exudation, by 3-fold (*P* < 0.001). However, in trees that were exposed to drought and then re-irrigated, this pattern reversed, and inoculated trees had slightly lower exudates than without bacteria. Thus, only under drought conditions were trees responsive to inoculation, showing a ~50% increase in root exudation (Figure 2; *P* = 0.039 for the interaction irrigation:bacteria)”.

L174. How is a correlation coefficient calculated when you have one root system (which are used to generate exudation data) but both types of bacteria? Please explain what variables are being used.

Indeed, we had a single exudate mean per treatment and time, and two bacterial abundance values, one for each of the two species. While interactions between the populations of the two strains may exist, it is still possible to correlate between each population and the exudate rates.

L186. Figure 5B?L202. Is this meant to be Figure 5B (and not 4B)?

Figure numbers were corrected.

L290. Replace "C source" with "C uptake" or "C assimilation"?

Text was changed: “Despite the lower carbon uptake, trees continued to exude altered metabolite blends”.

L298. But lower assimilation in the bacteria-drought treatment (as least based on Figure S2).

Text was added for clarity: “In turn, drought trees that were inoculated had increased biomass increment, as well as leaf nutrients and slightly higher photosynthetic activity before re-irrigation (Figure 5a, Figure S3, S4)”.

L311. Replace "sustain" with "remain"?

Done.

L365. The was these references are used in this sentence makes it seem like the studies were of tree roots (but they were not).

We thank the reviewer for the important point. The references were replaced with a more relevant citation: “Tree roots exude a plethora of secondary metabolites into the rhizosphere, which aid in the mobilization and uptake of essential macro-elements as N, P, and microelements like Mg, Mn, Zn, and Fe (Michalet et al. 2013)”.

Michalet, Serge, Julien Rohr, Denis Warshan, Clément Bardon, Jean-Christophe Roggy, Anne-Marie Domenach, Sonia Czarnes et al. "Phytochemical analysis of mature tree root exudates in situ and their role in shaping soil microbial communities in relation to tree Nacquisition strategy." *Plant physiology and biochemistry* 72 (2013): 169-177.

L486-489. Was any data collected on the changes in PAR over the course of the experiment? Were trees in full sun?

Information was added per the reviewer’s question: “Global solar radiation was 300-350 W m^-2^, reduced by ~15% by the net-house”.

L633. Do you have data on what C contamination existed in the non-root controls? Was this subtracted from the values of the ones with roots (i.e. were the fluxes corrected for background interferences)? Some info in the challenges of getting a C-free root chamber would be helpful to readers who want to repeat these methods.

Point well taken. Information was added in the text: “Two samples and one control (solution subjected to the same process without a root; metabolites that were found in control samples were not determined as exudate metabolites) per Rhizobox were included. Small carbon amounts that were found in the control (root-free) tubes were regarded as contamination and were subtracted from the carbon amounts in the samples”. C contamination was technical and resulted from the use of polypropylene syringes to collect the exudates. We assume that the level of contamination depends on the type of syringe and the manufacturing company, so these should be tested and controlled for on a specific experiment basis.

[Editors’ note: what follows is the authors’ response to the second round of review.]

The manuscript has been improved but there are some remaining issues that need to be addressed, as outlined below:Essential revisions:1) eLife has a consultative review procedure. Subsequent to the submission of both reviews, we discussed Figure 2 due to the Editor's concerns. This figure presents data which are very important for interpretation of the authors' proposed mechanism. In the current form of the manuscript the interpretation of these data is misleading.(1a) These data show that drought-stressed trees release less TOC (exudate) from roots than irrigated trees, and that both drought-stressed and irrigated trees release more TOC when inoculated. In contrast, trees re-irrigated after drought produce similar TOC from roots whether they are inoculated or not (although the figure indicates there is also a significant difference there, which is confusing and perhaps incorrectly indicated). Thus, the inference is rather that inoculated plants produce more TOC from roots, except if they have been subjected to drought and re-irrigation. What the authors imply is that drought-stressed plants produce more exudates when inoculated, which is correct but misleading since irrigated plants do as well. It is rather interesting that re-irrigated plants do not do this. Reviewer 1 commented additionally: "This contrasts with the answer of the authors to my main question: 'Thus, only under drought conditions were trees responsive to inoculation, showing a ~50% increase in root exudation (Figure 2; P = 0.039 for the interaction irrigation:bacteria).' I really cannot see this interaction in Figure 2, and if anything it seems to go into the wrong direction (i.e. difference with-without bacterial is larger under irrigation)."

Thank you for this point. We understand that our text was still somewhat confusing, and hence revised it to clarify our arguments and interpretation of the data: “Under constant irrigation and under drought, bacterial inoculation significantly increased root exudation, by 2-3-fold (Figure 2; *P* < 0.001). However, in trees that were exposed to drought and then re-irrigated, this pattern reversed, and inoculated trees had slightly lower exudates than without bacteria. Thus, drought-exposed trees, that were responsive to inoculation, showing a ~50% increase in root exudation, lost this response when re-irrigated (*P* = 0.039 for the interaction irrigation:bacteria). The amount of organic carbon exuded from roots of drought-exposed trees was significantly, ~50% lower than from roots of irrigated trees, and moreover, ~70% lower, in the re-irrigation period (across inoculation treatments; *P* < 0.01; Figure 2, Source Data 3, Supplementary Information). Importantly, the same trees that were exposed to drought, reduced their exudation when re-irrigated (Figure 2)”. We also clarified the significance of our results in the section “*A dynamic rhizosphere interplay between tree roots and soil bacteria*” in the Discussion: “The higher root exudation during drought than under re-irrigation is not trivial, considering that wet soil provides a better environment than dry soil for both tree roots and bacteria development (Gao et al. 2021)”. Considering that the experiment was held in Israel in May-June, i.e. going into the summer, conditions were very dry and hot (Figure S1, S2), making our observations of higher root exudation in drought vs. re-irrigated trees all the more surprising.

Gao, D., Joseph, J., Werner, R. A., Brunner, I., Zürcher, A., Hug, C., … and Hagedorn, F. (2021). Drought alters the carbon footprint of trees in soils—tracking the spatio‐temporal fate of 13C‐labelled assimilates in the soil of an old‐growth pine forest. *Global Change Biology*, *27*(11), 2491-2506.

(1b) The exudate production has no predictable correlation with abundance of either of the two bacterial strains monitored by the authors. The reason for choosing these strains is now clear in the text, but there is no indication that either responds to differences in root exudates. Thus, although the growth of both strains is affected by (some of) the exudate metabolites which differ (mildly – less than 2x) in abundance between treatments, there is no evidence that these two strains are directly associated with exudate differences. The inoculation affects total TOC from roots, but the effect could be indirect in the ecologically relevant, but complex soil environment of the experiment.

We agree with the Editor that the data from the tree experiment cannot provide direct evidence of exudate-induced growth enhancement of bacteria. Such evidence comes from our bacterial growth essays, which were done in vitro, rather than in vivo (in terms of the tree involvement; Figure 4). We do see positive correlations between exudation rate and rhizosphere abundance of both strains under drought (on Day 3; Figure 2C), but not under irrigation (except for *Bacillus subtilis* on Day 7; Figure 2C), nor re-irrigation. Therefore, overall, these correlations were inconsistent over times and treatments. Per the Editor’s comment, we toned down our claim in the opening paragraph of the Discussion: “Soil bacteria associated with tree roots and grew preferentially in the rhizosphere (than in the bulk soil; Figure 1), where their presence transiently increased with root exudates (Figure 2, Figure 7)”. We also agree that the interactions between root exudation rate and bacteria could be indirect in the complex soil environment of the experiment. We acknowledge and expand on that in the Discussion section “*A dynamic rhizosphere interplay between tree roots and soil bacteria*”: “Our temporal resolution and complex experimental setup do not address the question if exudates attracted the bacteria, or, alternatively, whether the bacteria induced root exudation. It is likely that both processes occurred simultaneously. One must also consider alternative explanations, e.g. that exudates act directly on soil elements, rather than solicited by bacteria. During drought, changes in soil water content could become either beneficial (increased Mn and P availability) or harmful (decreased Zn availability) to plant nutrition (Misra and Tyler 1999). Tree roots exude a plethora of secondary metabolites into the rhizosphere, which aid in the mobilization and uptake of essential macro-elements as N, P, and microelements like Mg, Mn, Zn, and Fe (Michalet et al. 2013). Several studies reported that beneficial rhizosphere bacteria drive an accumulation of elements (Philippot *et al.,*2013; Igiehon *et al.,* 2018). Although we showed that multiple exudate metabolites promoted bacterial growth, this does not mean that all of these metabolites act as specific cues for specific bacterial strains in the rhizosphere. However, the intimate attachment of bacteria to roots shown here should ensure the harvest of these metabolites by bacteria”. Finally, we refer to this option in the Discussion section on study limitations: “We offered a glimpse into the rhizosphere, which was not, however, free of limitations. First, our approach ignored the native rhizosphere microbiome, which probably affected many of the studied parameters”.

(1c) There are no data to indicate that these bacterial strains are responsible for the nutrient effects observed for inoculated plants undergoing drought, especially as the soil for the experiment was not sterilized. The unsterilized soil is in fact a nice aspect of the design from an ecological perspective, but also makes it more complicated to determine the mechanism by which additional inoculation has its effects.

Point well taken. As for the previous comment, we toned down our claim. This affected text in the Results: “Levels of P, Fe and Zn were significantly lower in leaves in drought compared to irrigated trees during the middle of the drought period (27.05). This effect was mitigated in inoculated trees under drought for P and Fe, but not for Zn (Figure 5A, Source Data 6)”. Please note that the Discussion text is already cautious with this matter, avoiding bald claims of a direct effect of the bacterial strains on leaf elements, e.g. in the first Dicussion paragraph: “In turn, drought trees that were inoculated had increased biomass increment, as well as leaf nutrients and slightly higher photosynthetic activity before re-irrigation (Figure 5a, Figure S3, S4)”, and in the third Discussion paragraph: “Finally, desiccated trees supplemented with soil bacteria showed better nutrition in P and Fe (Figure 5), and higher biomass (although not significantly)”. As detailed above, we acknowledge the role of the existing rhizosphere microbiome in our experiment, including its restrictions on our ability to draw direct evidence about the effects of the chosen strains. However, we do believe that the observed effects on leaf elements can be ascribed to our bacterial inoculations, for the following reasons: (1) When comparing the responses of inoculated drought-exposed trees to those without supplemented bacteria, a significant enhancement emerged. (2) The bacterial inoculations were most probably at higher numbers than the existing bacterial populations (as explained in the text), and hence must have had larger effects than the latter. (3) At least for P nutrition, our soil phosphorous measurements seem to support and complement the beneficial effect of inoculation on drought-exposed trees. (4) Previous studies have shown the beneficial effects of these strains to plants, e.g. via stimulating iron acquisition (Mendes et al. 2013 cited in text) suppressing plant pathogens (Weller et al. 2002, Raaijmakers et al. 2010 cited in text), and specifically fungi (Mendes et al. 2013). Finally, to provide a direct link between these bacterial strains and nutrition benefits to trees, we assume that additional approaches should be applied, e.g. growing plants in hydroponic system, and applying isotopic labeling to track specific elements as they transfer from roots to leaves.

(2) Reviewer 1 emphasized that the writing still needs substantial correction and streamlining.

Following this comment, an English editor has reviewed once again the entire text, correcting the remaining typos and grammar mistakes.

The full reviews are provided below. Please note that the topics under (1) arose after the reviews were submitted and during the consultation.Reviewer #1 (Recommendations for the authors):This manuscript has been improved during revision, including improved statistical analysis. In my view the Discussion would still benefit from a better focus on the main results.I mentioned in the previous review that parts of the paper were quite long and complicated to read and still find this to be true (see e.g. Discussion lines 413-443; overall, the Discussion has become even longer than it was before). I also mentioned that there were many small language errors and again can still find them in this "revised" version. It is really the authors responsibility to correct the language because they are the "publisher".

We thank Reviewer 1 for the valuable inputs. The Discussion is again longer than expected, due to journal’s regulations to avoid Discussion chapters in Supplementary Information. However, it has been condensed as much as possible, while avoiding compromising the wealth of insights arising from such an experiment. Per the reviewer’s comment we were able to mildly shorten the Discussion paragraphs under the section “*The chemistry of tree–microbe interactions in the rhizosphere*”, and in a few other places in text. Finally, an English editor has reviewed once again the entire text, correcting the remaining typos and grammar mistakes.